

**Evaporation measurement and modelling of an alpine saline lake influenced by freeze–thaw on the**
**Qinghai–Tibet Plateau**
Fangzhong Shi[1,2], Xiaoyan Li[1,2,3,4], Shaojie Zhao[1,2], Yujun Ma[5], Junqi Wei[1,2], Qiwen Liao[1,2], Deliang
Chen[6]
[1]State Key Laboratory of Earth Surface Processes and Resource Ecology, Faculty of Geographical
Science, Beijing Normal University, Beijing 100875, China;
[2]School of Natural Resources, Faculty of Geographical Science, Beijing Normal University, Beijing
100875, China
[3]Key Laboratory of Tibetan Plateau Land Surface Processes and Ecological Conservation, Ministry of
Education, Qinghai Normal University, Xining, China
[4]Academy of Plateau Science and Sustainability, Qinghai Normal University, Xining, China
[5]School of Geography and Planning, Sun Yat–sen University, Guangzhou, China
[6]Regional Climate Group, Department of Earth Sciences, University of Gothenburg, Gothenburg,
Sweden.
* To whom the correspondence should be addressed: Xiao–Yan Li, State Key Laboratory of Earth
Surface Processes and Resource Ecology, Beijing Normal University, Beijing, China. Emails:
xyli@bnu.edu.cn.
✉ **Fangzhong Shi (during review)**      **fzshi@mail.bnu.edu.cn**
✉ **Xiaoyan Li* (after acceptance)**      **xyli@bnu.edu.cn**





**Key Points**

- Night evaporation of Qinghai Lake accounts for more than 40% of the daily evaporation during both the ice–free and ice–covered periods.

- Lake ice sublimation reaches 175.22±45.98 mm, accounting for 23% of the annual evaporation.

- Wind speed weakening may have resulted in an 11.14% decrease in lake evaporation during the ice–covered period from 2003 to 2017.





## Abstract

Saline lakes on the Qinghai–Tibet Plateau (QTP) profoundly affect the regional climate and water cycle
through loss of water (E, evaporation under ice–free (IF) and sublimation under ice–covered (IC)
conditions). Due to the observation difficulty over lakes, E and its underlying driving forces are seldom
studied targeting saline lakes on the QTP, particularly during the IC. In this study, E of Qinghai Lake
(QHL) and its influencing factors during the IF and IC were first quantified based on six years of
observations. Subsequently, two models were chosen and applied in simulating E and its response to
climate variation during the IF and IC from 2003 to 2017. The annual E sum of QHL is $768.58 \pm 28.73$
mm, and E sum during the IC reaches $175.22 \pm 45.98$ mm, accounting for 23% of the annual E sum. The
E is mainly controlled by the wind speed, vapor pressure difference, and air pressure during the IF, but
driven by the net radiation, the difference between the air and lake surface temperatures, wind speed, and
ice coverage during the IC. The mass transfer model simulates lake E well during the IF, and the model
based on energy achieves a good simulation during the IC. Moreover, wind speed weakening results in
an 11.14% decrease in E during the IC of 2003–2017. Our results highlight the importance of E in IC,
provide new insights into saline lake E in alpine regions, and can be used as a reference to further improve
hydrological models of alpine lakes.

## Keywords:

Lake evaporation and sublimation, saline lakes, flux observation, ice–covered period, Qinghai Lake,
Qinghai–Tibet Plateau



## 1. Introduction

Saline lakes account for 23% of the total area and 44% of the total water volume of Earth's lakes (Wurtsbaugh et al., 2017). They play an important role in shaping the regional climate and maintaining ecological security and sustainable development in arid regions (Messager et al., 2016; Wurtsbaugh et al., 2017; Woolway et al., 2020; Wu et al., 2021; Wu et al., 2022). Under the influences of climate change and human activities, saline lakes worldwide have changed rapidly in terms of their area, level, temperature, ice phenology, energy and water exchange, which has become an issue of concern (Gross, 2017; Wurtsbaugh et al., 2017; Woolway et al., 2020). Evaporation under ice–free (IF) and sublimation under ice–covered (IC) periods (E) is an important mechanism of transfer of energy and water between lakes and atmosphere, and is one of the main factors influencing changes in the lake water volume (Lazhu et al., 2016; Ma et al., 2016; Woolway et al., 2018; Guo et al., 2019; Woolway et al., 2020).

In contrast to freshwater lakes, E of saline lakes involves a more complex process and is affected not only by climate conditions but also by the salinity, lake depth, temperature, stratification, thermal stability, and hydrodynamics (Hamdani et al., 2018). For example, dissolved salt ions can reduce the free energy of water molecules (i.e., reduced water activity) and result in a reduced saturated vapor pressure above saline lakes at a given water temperature (Salhotra et al., 1987; Mor et al., 2018). Previous studies have investigated the relationship between E and salinity of saline lakes and discrepancies in the controlling factors between different time scales (Salhotra et al., 1987; Lensky et al., 2018; Hamdani et al., 2018; Mor et al., 2018). These studies have mainly focused on saline lakes in arid and temperate zones, and the interaction and mutual feedback between the water body of saline lakes and the atmosphere remain unclear. In particular, there are few studies on E of alpine saline lakes which exhibit complex hydrology and limnology.

Saline lakes account for over 70% of the total lake area on the Qinghai–Tibet Plateau (QTP) (Liu et al., 2021), and thus profoundly affect the regional climate and water cycle through E (Yang et al., 2021). However, continuous year–round direct measurements of saline lake E are scarce, which hinders the exploration of lake E at different time scales. Observations of E from saline lakes have been obtained for Qinghai Lake (QHL) (Li et al., 2016), Namco (Wang et al., 2015; Ma et al., 2016), Selinco (Guo et al., 2016), and Erhai (Liu et al., 2015) via the eddy–covariance (EC) technique or pan E on the QTP, but


these observations are mainly for the growing seasons (or IF: approximately mid–May to mid–October).
Thus, there are considerably fewer E observations during the IC and full–year period of lakes, mainly
because of the harsh environment and limited accessibility to the QTP (Lazhu et al., 2016). However,
most lakes on the QTP exhibit a long and stable IC lasting more than 100 days due to the low annual air
temperature (Ta) (Cai et al., 2019), which suggests that E observations are currently lacking for nearly a
quarter of the year (from the IF to the IC). Although studies have commented on the importance of E
during the IC (Li et al., 2016; Wang et al., 2020) and clarified that freezing/breakup processes could result
in sudden changes in lake surface properties (such as albedo and roughness) and affect the water and
energy exchange between the lake and atmosphere (Cai et al., 2019; Yang et al., 2021), the dynamic
processes of energy interchange and E of saline lakes during the IC and its responses to climate warming
on the QTP still constitute a knowledge gap in lake hydrology research. Thus, there is an urgent need to
better quantify lake E during the IC on the QTP.
A large number of models have been employed to calculate lake E, mainly including the Dalton formula
series based on mass transfer and aerodynamics, energy and water balance formula series, Penman
formula series considering both aerodynamics and energy balance, and empirical formula based on
statistical analysis (Dalton, 1802; Bowen, 1926; Penman, 1948; Harbeck et al., 1958; Finch and Calver,
2008; Hamdani et al., 2018; Wang et al., 2019a). However, the reported values exhibit large discrepancies
in their seasonal variations and annual amounts between those models (Lazhu et al., 2016; Ma et al.,
2016; Guo et al., 2019; Wang et al., 2019a; Wang et al., 2020)., and almost all models were calibrated
and verified against E observations during the IF as a result of the deficiency in observed E during the
IC (Lazhu et al., 2016; Guo et al., 2019), and E during the IC was either not calculated or unverified
(Wang et al., 2020). In addition, compared with small lakes, large and deep lakes exhibit higher E levels
and delayed seasonal E peaks because more energy is absorbed and stored in large and deep lakes during
the IF and released during the IC (Wang et al., 2019a). Thus, the effect of changes in ice phenology on
lake E is particularly important, which calls for different models for E simulation during the IF and IC.
Furthermore, with increasing overall surface air warming and moistening, solar dimming, and wind
stilling since the beginning of the 1980s (Yang et al., 2014), lakes on the QTP have experienced a
significant temperature increase (at a rate of 0.037°C/yr from 2001 to 2015) (Wan et al., 2018) and ice



phenology shortening (at a rate of −0.73 d/yr from 2001 to 2017) (Cai et al., 2019). Changes in the air
Ta, water surface temperature (Ts), wind speed (WS), and ice phenology could impose different effects
on energy interchange and molecular diffusion due to differences in the state phase and reflectance of
water between the IC and IF, thus altering lake E (Wang et al., 2018). Although many studies have
reported a decrease in E of lakes on the QTP by model simulations (Lazhu et al., 2016; Ma et al., 2016;
Li et al., 2017; Guo et al., 2019), owing to E neglect during the IC, the potential mechanisms of lake E
and its different responses to climate change during the IC and IF remain unclear.
In this study, based on six continuous years of direct measurements of lake E and energy exchange flux
data obtained with the EC technique pertaining to QHL, the largest saline lake on the QTP, between 2014
and 2019, we quantified the characteristics of energy interchange and E on diurnal, seasonal (IF, IC and
cycle year: AN) and yearly time scales and identified the potential influencing factors of E during the IF
and IC. In addition, combined with reanalysis climate datasets, a mass transfer model (MT model), and
a model based on energy, temperature and WS (JH model) were calibrated and verified, with the optimal
model chosen for the simulation of lake E and its response to climate change during the IF and IC from
2003 to 2017. The results would highlight the importance and potential mechanisms of E during IC, and
can be used as a reference to further improve hydrological models of alpine lakes.
**2. Materials and Methods**
**2.1. Site description and energy exchange flux and climate data**
QHL (36°32′−37°15′ N, 99°36′−100°47′ E, 3194 m a.s.l.), with an area of 4,432 km$^2$ and a catchment of
29,661 km$^2$, is the largest inland saline lake in China (Li et al., 2016). The average depth of the lake is
26 m. The average salt content is 14.13 g L$^{-1}$, and the pH ranges from 9.15 to 9.30. The hydrochemical
type of the lake water is Na–SO4–Cl (Li et al., 2016). The mean annual Ta, precipitation, and E values
between 1960 and 2015 were −0.1°C, 355 mm and 925 mm, respectively (Li et al., 2016). The IC usually
begins in late November, ends in mid–late March or even early April, and lasts more than 100 days.
Under the effects of climate warming, QHL has experienced temperature increases, area expansion, and
IC shortening.
The instruments to measure the energy exchange flux and micrometeorological parameters were installed





---

at the China Torpedo Qinghai Lake test base (36°35′27.65″ N, 100°30′06″ E, 3198 m a.s.l.) located in
the southeastern QHL approximately 737 m from the nearest shore (Li et al., 2016) (Fig. 1). The water
depth underneath this platform is 18 m. The torpedo test tower has a height of 10 m above the water
surface. The EC system was installed on a steel pillar mounted on the northwestern side of the top of the
torpedo test tower with a total height of 17.3 m above the lake water surface (Li et al., 2016). A three–
dimensional sonic anemometer (model CSAT3, Campbell Scientific Inc., Logan, UT, USA) was used to
directly measure horizontal and vertical wind velocity components (u, v, and w) and virtual temperature.
An open–path infrared gas analyzer (model EC150, Campbell Scientific Inc.) was applied to measure
fluctuations in water vapor and carbon dioxide concentrations. Fluxes of sensible heat (H) and latent heat
(LE) were calculated from the 10–Hz time series at 30–min intervals and recorded by a data logger
(CR3000, Campbell Scientific Inc.). The observation instruments were powered by solar energy.
A suite of auxiliary micrometeorology was also measured as 30–min averages of 1–s readings on the
eastern side of the top of the torpedo test tower, 3 m away from the EC instruments. The net radiation
(Rn) was calculated from the incoming shortwave, reflected shortwave, and incoming and outgoing
longwave radiation, which were measured by a net radiometer (CNR4, Kipp & Zonen B.V., Delft,
Netherlands) at 10 m above the lake surface (Fig. 1). The Ta, relative humidity (RH) and air pressure
(Pres) were measured at a height of 12.5 m above the water surface. A wind sentry unit (model 05103,
RM Young, Inc. Traverse City, MI, USA) was employed to measure the WS and wind direction (WD).
The Ts was measured with an infrared thermometer (model SI–111, Campbell Scientific Inc.)
approximately 10 m above the water surface, and the water temperature (Tl) was measured with a
temperature probe (109 L, Campbell Scientific Inc.) at depths of 0.2, 0.5, 1.0, 2.0 and 3.0 m. precipitation
was measured with an automated tipping–bucket rain gauge (model TE525, Campbell Scientific Inc.)
and precipitation gauge (model T–200B, Campbell Scientific Inc.). The observation system began
operation on May 11, 2013. In this study, we unified all observational data at 30–min intervals and
analyzed the data from January 1, 2014 to December 31, 2019.

**2.2. Reanalysis climate datasets**

The reanalysis climate datasets used to drive the lake E models were acquired from the interim reanalysis
dataset v5 (ERA5) produced by the European Centre for Medium–Range Weather Forecasts





(https://cds.climate.copernicus.eu/cdsapp#!/search?type=dataset) and the China Regional High–
Temporal–Resolution Surface Meteorological Elements–Driven Dataset (CMFD)
(http://data.tpdc.ac.cn/en/). Gridded hourly ERA5 skin temperature and daily WS, daily CMFD Ta, Pres,
RH, and downward shortwave radiation (Rs) at a spatial resolution of 0.1° from 2001 to 2018 were
analyzed in this study. The daily skin temperature was generated by averaging the hourly temperature
over 24 h per day and was adopted as the lake surface temperature. We extracted climate data pertaining
to QHL via a grid mask with a spatial resolution of 0.1° and averaged the data in all pixels. Considering
the advantages of long time spans and high resolution, the ERA5 and CMFD datasets developed based
on land station data have been recognized as the best currently available reanalysis products and have
been widely applied in land–surface and hydrological modelling studies in China (Lazhu et al., 2016; Ma
et al., 2016; Tian et al., 2021; Xiao and Cui, 2021). To reduce the uncertainty caused by the input data,
the daily lake surface temperature from EAR5, Ta and Rs from CMFD for QHL were adjusted with fitting
equations of the observed daily Ts ($R^2 = 0.92$, $P < 0.01$), Ta ($R^2 = 0.90$, $P < 0.01$) and Rs ($R^2 = 0.73$, $P <$
0.01) from 2014 to 2018 (Fig. S1), and the equations were shown as below:
$$T_a^{ad} = 1.01 \times T_a^{CMFD} + 0.71 \tag{1}$$
$$T_s^{ad} = 0.71 \times T_s^{ERA5} + 3.30 \tag{2}$$
$$R_s^{ad} = 0.86 \times R_s^{CMFD} + 34.63 \tag{3}$$
where $T_a^{ad}$, $T_s^{ad}$ and $R_s^{ad}$ are Ta, Ts and Rs, respectively, after adjustment.
**2.3. Lake ice coverage dataset and ice phenology**
The daily lake ice coverage of QHL from 2002 to 2018 was extracted from a lake ice coverage dataset
of 308 lakes (with an area greater than 3 km$^2$) on the QTP retrieved from the National Tibetan Plateau
Data Center (http://data.tpdc.ac.cn/en/). The dataset with a time span from 2002 to 2018 was generated
from the Moderate Resolution Imaging Spectroradiometer (MODIS) normalized difference snow index
(NDSI) product with the SNOWMAP algorithm, and the data under cloud cover conditions were
redetermined based on the temporal and spatial continuity of lake surface conditions (Qiu et al., 2019).
Based on the lake ice coverage, the IF was defined as an ice coverage lower than 10%, and the IC was
defined as an ice coverage higher than 10% (Qiu et al., 2019). The IC was divided into three stages:





freeze (FZ: 10% < ice coverage < 90%), completely freeze (CF: ice coverage > 90%) and thaw (TW: 10%
< ice coverage < 90%) (Qiu et al., 2019). We defined the cycle year (annual: AN) from the beginning of
the IF to the end of the IC.
**2.4 Data processing of the observed energy exchange flux and climate data**
The EC fluxes were processed and corrected based on the 10–Hz raw time series data in the data
processing software EdiRe, including spike removal, lag correction of water to carbon dioxide relative
to the vertical wind component, sonic virtual temperature correction, performance of planar fit coordinate
rotation, density fluctuation correction (WPL correction) and frequency response correction (Li et al.,
2016). Since the shortest distance between the Chinese torpedo Qinghai Lake test base and the
southwestern lakeshore is only 737 m, there may be insufficient fetch for a turbulent flux under certain
conditions. Therefore, footprint analysis was conducted to eliminate data influenced by the surrounding
land. For further details on the process and results of the footprint analysis, see Li et al. (2016). In addition
to these processing steps, quality control of the 30–min flux data was conducted using a five–step
procedure: (i) data originating from periods of sensor malfunction were rejected (e.g., when there was a
faulty diagnostic signal), (ii) data within 1 h before or after precipitation were rejected, (iii) incomplete
30–min data were rejected when the missing data constituted more than 3% of the 30–min raw record,
(iv) data were rejected at night when the friction velocity was below 0.1 m/s (Blanken et al., 1998) and
(v) data with large footprints (>700 m) and a wind direction from 180° to 245° were eliminated.
To further control the quality of the energy exchange flux (sensible heat flux and latent heat flux: H and
LE, respectively) and micrometeorological dataset (Rn, Ta, Ts, Tl, RH, WS, Pres, and albedo), data
outside the mean ± 3 × standard deviation were removed for each variable. Then, gap–filling methods
entailing a look–up table and mean diurnal variation (Falge et al., 2001) were adopted to fill gaps in the
flux measurement data. The look–up table method was applied when the meteorological dataset was
available synchronously. Otherwise, the mean diurnal variation method was adopted. The heat storage
change (G, W/m$^2$) was estimated as a residual of the energy balance:
$G = Rn - LE - H$ (4)
where Rn is the net radiation (W/m$^2$), H is the sensible heat flux (W/m$^2$) and LE is the latent heat flux
(W/m$^2$). Lake E was calculated as





$E = \lambda \times LE$ (5)
where $\lambda$ is the latent heat of vaporization (MJ/kg), taken as 2.45 MJ/kg in this paper (Allen et al.,

216    1998).

**2.5. Models for daily lake evaporation simulation**
To evaluate the interannual variation in QHL E from 2003 to 2017, we validated three models during the
AN, IF, and IC periods. The three models were as follows:
1) Mass–transfer model (MT model) (Harbeck et al., 1958)
$E_{MT} = N \times F(WS) \times \Delta e$ (6)
$F(WS) = a1 \times WS + a2$ (7)
$\Delta e = \alpha \times e_s - e_a$ (8)
$e_s = 6.105 \times \exp \left(\frac{17.27 \times Ts}{Ts+237.7}\right)$ (9)
$e_a = 6.105 \times \exp \left(\frac{17.27 \times Ta}{Ta+237.7}\right)$ (10)
where $E_{MT}$ is the E rate (mm/day); N is the mass–transfer coefficient; WS is the wind speed (m/s); $\Delta e$
is the vapor pressure difference and $\alpha$ is the water activity coefficient for saline lakes, which represents
the ratio between the vapor pressure above saline water and that above freshwater at the same temperature,
and an $\alpha$ value of 0.97 was suggested for QHL, as measured with a portable water activity meter
(AwTester, China). Moreover, $e_s$ and $e_a$ are the saturated vapor pressure at the lake surface
temperature (Ts) and air temperature (Ta), respectively. This model inherently accounts for the water
salinity through $\Delta e$ and requires calibration of coefficients N, a1 and a2, which were taken as 1.86, 0.01,
and 0.13, respectively, during the AN; 0.18, 0.40, and 0.26, respectively, during the IF; and 0.97, 0.09,,
and 0.38, respectively, during the IC in this paper.
2) Atmospheric dynamics model (AD model) (Hamdani et al., 2018)
$E_{AD} = \frac{0.622 \times Ce}{\rho_w \times P} \times \rho_a \times WS \times 3.6 \times 10^6 \times \Delta e$ (11)



$$\rho_a = 1.293 \times (\frac{273.15}{273.15+Ta}) \times \frac{Pres}{101.325}$$ (12)
where $\rho_w$ and $\rho_a$ denote the water and air densities (kg/m³), respectively, and $\rho_w$ is approximately
$1.011 \times 10^3$ for QHL. Moreover, $Pres$ is the air pressure (mbar), and $Ce$ is a transport coefficient
obtained via calibration to address missing friction velocity values in the reanalysis climate datasets,
which was taken as $3.20 \times 10^{-3}$, $3.00 \times 10^{-3}$ and $6.60 \times 10^{-3}$ during the AN, IF and IC, respectively, in
this paper.
3) Statistical model based on solar radiation (the Jensen–Haise method: JH model) (Wang et al., 2019a)
$$E_{JH} = JH1 \times (JH2 \times (Ta - Ts) + JH3) \times (Rs) \times (WS)$$ (13)
where Rs is the incoming solar shortwave radiation (W/m²); JH1, JH2 and JH3 must be calibrated and
were taken as $6.80 \times 10^{-3}$, $-0.01$ and $0.38$, respectively, during the AN; $0.03$, $-3.80 \times 10^{-3}$ and $0.08$,
respectively, during the IF; and $6.90 \times 10^{-3}$, $0.02$ and $0.49$, respectively, during the IC in this paper.
The three models were selected, firstly as they are typical representatives in considering mass transfer,
aerodynamics, energy transfer, respectively; secondly because their demand parameters are easy to
acquire, which are adaptive to be promoted; and third as they have been proved to be efficient in saline
lakes (Hamdani et al., 2018). These models were first calibrated and validated based on daily E
observations from 2014 to 2019 during the different periods of the AN, IF and IC. The root–mean–square
error (RMSE) and goodness of fit ($R^2$) were used to evaluate the effectiveness of the models. A model
with high $R^2$ and low RMSE values was selected for lake E simulation during the AN, IF and IC periods.
**2.6. Statistical analysis**
Summer and autumn were taken as June to August and September to November, respectively. During
data analysis, we first divided the 30–min observed energy exchange flux and climate data from 2014 to
2019 by the AN, IF, and IC based on the calculated ice phenology. Hence, we obtained datasets of five
cycle years from the IF in 2014 to the IC in 2018 (Fig. S2). Second, we calculated the multiday average
30–min observed energy exchange flux during the IF and IC in each year to evaluate the basic statistical
characteristics of the diurnal E and exchange flux. The daily energy exchange flux and climate data were
then calculated by averaging the 30–min data for each day, and one–way ANOVA was performed to





compare the difference in E and G between the IF and IC in each year from 2014 to 2018. Third, to
explore the key factor controlling lake E, partial least squares regression and random forest methods were
used to calculate the sensitivity coefficient (standing for the regression coefficient of each variable, which
means the amount of change in E caused by the variation of per unit in the variable) and importance of
Rn, WS, Δe, Pres, albedo, WD, Ta−Ts, Tl, and ICR to E during the daytime and nighttime IF and IC,
respectively. Finally, three models were validated and two models were selected to severally calculate
the interannual E during the IF and IC from 2003 to 2017 (the available ice phenology exhibits a limited
cycle year from 2003 to 2017). Four controlled tests were then conducted to quantify the contribution of
the variation in Ta, Ts, WS, and Rs to lake E from 2003 to 2017. The partial least squares and random
forest analyses were conducted in R and the other analyses were conducted in MATLAB.
**3. Results**
**3.1. Diurnal and seasonal characteristics of evaporation and the energy budget during the different**
**freeze–thaw periods**
The average E, LE, G, H, and Rn values (average from 2014 to 2018) were $1.20 \pm 0.09$ mm/d, $68.01 \pm$
$4.93$ W/m$^2$, $192.18 \pm 7.00$ W/m$^2$, $16.25 \pm 1.21$ W/m$^2$ and $276.45 \pm 3.32$ W/m$^2$, respectively, during the
IF; and $1.11 \pm 0.20$ mm/d, $63.15 \pm 11.31$ W/m$^2$, $79.23 \pm 18.12$ W/m$^2$, $4.68 \pm 0.37$ W/m$^2$ and $147.06 \pm$
$14.23$ W/m$^2$, respectively, during the IC. The daytime E, LE, G, H and Rn values were notably lower
during the IC than those during the IF, except E and LE in 2014 (Figs. 2 and 3; Table S1). In addition,
the daily peak LE and E values typically occurred at approximately 12 pm during the IF and
approximately 2 pm during the IC, and exhibited an approximately two–hour lag during the IF and a
four–hour lag during the IC over G and Rn (Fig. 2). At night, although lower E (at an average rate of 0.81
$\pm 0.17$ mm/d) and LE ($46.02 \pm 9.71$ W/m$^2$) levels occurred during the IC than during the IF (at average
rates of $0.94 \pm 0.05$ mm/d and $53.09 \pm 2.94$ W/m$^2$, respectively), E (LE) accounted for 42%−45% and
41%−45% of the total daily E during the IF and IC, respectively (Figs. 2 and 3; Table S1). In regard to
G, a similar release rate was found during the IF and IC, but the heat release time was longer during the
IC than that during the IF (Fig. 2).
The daily E ranged from 1.96 to 2.34 mm/d during the IF and from 1.57 to 2.71 mm/d during the IC, and



the average E sum reached 593.37 ± 44.87 mm/yr during the IF and 175.22 ± 45.98 mm/yr during the IC
from 2014 to 2018 (Fig. 3; Fig. S2; Table S1). This suggests an average E sum of 77% during the IF and
23% during the IC throughout the cycle year from 2014 to 2018 (with a lake E sum ranging from 719.45
to 798.55 mm/yr and an average value of 768.58 ± 28.73 mm/yr) (Fig. 3). In terms of G, QHL initially
released heat in autumn, which lasted until the lake was completely frozen, after which heat was absorbed
from the lake thawing period throughout the summer (Fig. S2; Fig. S3).
**3.2. Response of evaporation to climatic factors during the different freeze–thaw periods**
The key controlling factor of lake E was explored based on the daily observed energy exchange flux and
climate data (E, Rn, WS, Δe, Pres, albedo, WD, Ta−Ts, and Tl) and ICR during the IF and IC from 2014
to 2018. The Δe (with a sensitivity coefficient of 0.28 in the daytime and 0.22 in the nighttime, $P < 0.05$),
WS (with a sensitivity coefficient of 0.54 in the daytime and 0.43 in the nighttime, $P < 0.05$) and Pres
(with a sensitivity coefficient of 0.26 in the daytime and 0.14 in the nighttime, $P < 0.05$) notably increased
E (Fig. 4), and the effect was greater in the daytime than that in the nighttime during the IF (Fig. 4). The
Rn (with a sensitivity coefficient 0.25 in the nighttime, $P < 0.05$), WS (with a sensitivity coefficient of
0.30 in the daytime and 0.22 in the nighttime, $P < 0.05$), Ta−Ts (with a sensitivity coefficient of 0.59 in
the daytime and 0.39 in the nighttime, $P < 0.05$) and ICR (with a sensitivity coefficient of 0.20 in the
daytime and 0.17 in the nighttime, $P < 0.05$) imposed a significant positive effect on E during the IC (Fig.
4). Similarly, the top five important factors calculated with the random forest method were WS, Δe, Pres,
WD, and Ts during the IF and Ta−Ts, Ta, WS, Rn, and ICR during the IC (Fig. S4). This indicates that E
of QHL is mainly controlled by WS, Δe, and Pres during the IF but is driven by Rn, Ta−Ts, WS, and ICR
during the IC.
**3.3. Evaporation simulation and interannual variation**
Three models (MT, AD, and JH) were calibrated and validated to evaluate the interannual variation in
QHL E from 2003 to 2017. In the case of model performance, the MT model based on molecular diffusion
performed the best in terms of E simulation during the IF (with the largest $R^2$ and smallest RMSE values
of 0.77 and 0.88, respectively), while the JH model based on energy exchange performed the best during
the IC (with the largest $R^2$ and smallest RMSE values of 0.68 and 1.07, respectively) (Figs. S5 and S6).
Thus, the interannual variation in QHL E from 2003 to 2017 was calculated with the MT model during





the IF and with the JH model during the IC (Fig. 5). From 2003 to 2017, although Ta (at a rate of
−0.01°C/yr), Pres (at a rate of −0.01 hPa/yr) and WS (at a rate of −0.006 m/(s·yr)) decreased, increases
in Δe (at a rate of 0.01 hPa/yr) and Ts (at a rate of 0.001°C/yr) resulted in an increase in E (at a rate of
1.49 mm/yr for the E sum) during the IF (Figs. 5 and S7). Conversely, ignoring the increases in Ta (at a
rate of 0.04°C/yr) and Ta−Ts (at a rate of 0.04°C/yr), with decreasing WS (at a rate of −0.008 m/(s·yr)),
E (at a rate of −1.96 mm/yr for the E sum) decreased during the IC, which resulted in an inapparent
decrease in E (at a rate of −0.46 mm/yr for the E sum) during the AN (Figs. 5 and S7).
**4. Discussion**
**4.1. Lake evaporation during the ice–covered period**
The results of this study highlight the important contribution of lake ice sublimation to the total amount
of lake E. Due to the low snow coverage of Qinghai Lake in winter (with a maximal snow coverage less
than 16% of the area of Qinghai Lake), evaporation and sublimation of lake ice and water are the main
sources of E during the IC of 2013~2018 (Fig S8). In liquid drops, E can be explained based on the
coffee–stain effect in which the local diffusion–limited E rate diverges at the contact line (the border of
the liquid drops) and outward flow from a given droplet replenishes the corner region if the droplet
contact line remains fixed (Deegan et al., 1997 and 2000). Similarly, ice crystal E also starts at the contact
line first and quickly recedes along sharp crystal edges (Nelson, 1998; Jambon–Puillet et al., 2018). Since
the mass loss caused by E cannot be replaced, the occurrence of E at sharp points causes these points to
successively retreat, resulting in self–similar smoothness (Jambon–Puillet et al., 2018). The experimental
and simulation results of Jambon–Puillet et al. (2018) verified that the E rates of liquid droplets and ice
crystals remain the same under unchanged environmental conditions. In this study, the E rate of QHL
during the IC ranged from 1.57 to 2.71 mm/d, approximately 0.73–1.38 times that of liquid water during
the IF (Table S1), with similar results to those findings of liquid droplets and ice crystals.
In practice, lake E varies diurnally, seasonally, and interannually with climatic and environmental
changes, and the E rate varies considerably among lakes in different regions. Few studies have examined
lake ice E during the IC, and most studies have focused on polar sea ice and alpine snow packs (Froyland
et al., 2010; Froyland, 2013; Herrero et al., 2016; Christner et al., 2017; Lin et al., 2020). Observational


and modelling studies of Antarctic ice sheets or lakes have found that the monthly E rate of ice ranged
from −4.6 to 13 mm/month from June to September (Antarctic) (Froyland et al., 2010). In this study, we
found that E sum ranges from 130.59 to 262.45 mm during the IC from 2014 to 2018, which is higher
than the previous observations from Antarctic ice sheets or lakes. This may be because Antarctic ice
sheets or lakes are located at high latitudes with low solar radiation and are therefore cooler from the
surface to greater depths with energy–limiting conditions for E (Persson et al., 2002). However, the lakes
on the QTP freeze seasonally, so most of these lakes can store a large amount of heat because of the high
solar radiation during the IF (Fig. 6), which could lead to the observed E during the IC (Huang et al.,
2011 and 2016). Studies on surface E of a shallow thermokarst lake in the central QTP region have found
that E reaches up to 250 mm/yr during the IC (Huang et al., 2016), which is close to our observed E
levels (130.59−262.45 mm/yr). Our results further showed that E of QHL accounted for 23% of the
annual E during the IC. Wang et al. (2020) evaluated 75 large lakes on the QTP and demonstrated that E
of these lakes in winter accounted for 12.3%−23.5% of the annual E, which suggests that E of these lakes
during the IC was the same as that during the other seasons. Furthermore, considering that the area of
QHL is 4,432 km$^2$ (Li et al., 2016), the QHL releases 3.39 ± 0.13 km$^3$ of water into the air every year,
which corresponds to the sum of the water for animal husbandry, industrial and domestic uses in Qinghai
province (an average of 2014 to 2017) (Dong et al., 2021).
**4.2. Responses of lake evaporation to salinity and climate change**
The salinity has a powerful influence on E of saline lakes by changing both water density and thermal
property, dissolved salt ions can reduce the free energy of water molecules, and result in a higher boiling
point and reduced saturated vapor pressure above saline lakes at a given water temperature (Salhotra et
al., 1987; Abdelrady, 2013; Mor et al., 2018). Therefore, increase in salinity of lake would decrease its E
rate. For example, Lee (1927) compared the E of pure water with that of saline lakes of different densities
(salinity) in Nevada, USA, and found that when the densities (salinity) of water increased by 1%, the E
of saline lake decreased by 0.01% compared with pure water. Similarly, Mor et al. (2018) found that E
rate of diluted plume is nearly three times larger than that in open lake in Dead Sea. Thus, the
thermodynamic concept of water activity (the water activity of freshwater is 1, while in saline water is
lower than 1, and the higher the salinity, the lower of water active in lakes.) has been widely used in E





simulations of saline lakes, which is defined as the ratio of water vapor pressure on the surface of saline
and fresh water at a given temperature (Salhotra et al., 1987; Abdelrady, 2013; Mor et al., 2018). In our
study, we measured the water activity of QHL was 0.97 by a salinity of 14.13 g $L^{-1}$, and applied it to the
models, which make it more theoretical to explain E process of saline lakes and reduced the uncertainty
of estimation in saline lake E. For example, with the salinity of 133 g $L^{-1}$ of surface water, water activity
was measured to be 0.65, and has been widely used in its E simulation of Dead sea (Metzger et al., 2018;
Mor et al., 2018; Lensky et al., 2018); and Abdelrady (2013) improved the surface energy balance system
(SEBS) of E in saline lakes by constructing an exponential function between lake salinity and water
activity, which reduced the simulated E by 27% and RMSE from 0.62 to 0.24 mm $3h^{-1}$ in the Great Salt
Lake. Therefore, considering salinity is very important to improve the accuracy of E simulation in saline
lakes. Certainly, lake salinity changes dynamically at diurnal, seasonal and interannual scales, but since
the difficult of continuously observation of lake salinity, a fixed water activity in our study may cause
the underestimate in E of QHL, due to the decrease of salinity by the expansion of QHL.
Furthermore, climate and environment are also important factors affecting lake E, and varied
significantly among the different seasons. Previous studies have shown that lake E is mainly affected by
WS and $\Delta e$ in summer and WS, $\Delta e$, Ta−Ts, and G in winter (Zhang and Liu, 2014; Hamdani, et al., 2018).
This suggests that energy exchange between lakes and air may be one of the main drivers of E during the
IC under the same atmospheric boundary conditions (Fig. 6). Since most lakes store heat in summer, they
release heat and sufficiently produce E in winter (Blanken et al., 2011; Hamdani, et al., 2018). In this
study, we also found that QHL began to store heat in the lake thawing period and released heat in autumn
or when the lake began to freeze (Figs. 6 and S3). Therefore, E of QHL was mainly controlled by WS,
$\Delta e$, and Pres during the IF, whereas it was mainly affected by Rn, Ta−Ts and WS during the IC (Fig. 6).
Considering the overall surface air warming and moistening, solar dimming, and wind stilling since the
beginning of the 1980s across the QTP (Yang et al., 2014), we further explored the contribution of Ta,
Ts, WS and Rs to E during the IF and IC from 2001 to 2017 by simulation tests. Compared with 2001, E
during the IC increased at an average rate (2003–2017) of $4.90 \pm 6.14$ mm/yr (3.52%) due to an increase
in Ta and decreased at an average rate (2003–2017) of $-5.84\pm3.54$ mm/yr (3.37%), $-6.17 \pm 4.77$ mm/yr
(3.19%), and $-18.92 \pm 27.55$ mm/yr (11.14%) due to an increase in Ts and decrease in Rs and WS,





respectively (Fig. 7; Table S2). Moreover, the increase in Ts increased E at an average rate (2003–2017)
of 10.19 ± 19.00 mm/yr (3.37%) during the IF (Fig. 7; Table S2). In addition, changes in lake ice
phenology significantly affected lake E during the IF and IC. Compared with 2003 to 2007 (101.40 ±
7.00 d), the average IC decreased by 10.8 d from 2013 to 2017 (90.60 ± 6.08 d) (Table S3). A shortened
IC suggests a much lower albedo in the cycle year and could result in higher Rs absorption and a shorter
period for heat–induced recession, which could increase lake E (Wang et al., 2018). Of course, lake E is
also affected by the lake area, water level, and physical and chemical properties (Woolway et al., 2020),
especially for saline lakes (Salhotra et al., 1987; Mohammed and Tarboton, 2012; Mor et al., 2018).
Increasing the water salinity could reduce E (Salhotra et al., 1987; Mor et al., 2018) because the dissolved
salt ions could reduce the free energy of water molecules (i.e., reduced water activity) and result in a
lower saturated vapor pressure above saline lakes at a given water temperature (Salhotra et al., 1987;
Mor et al., 2018). However, the changes in lake physical and chemical properties attributed to lake
freezing increase the complexity of the underlying mechanism, simulation of ice E and its response to
climate change, and more studies are needed to further explore interactions between the different factors.

### 4.3. Uncertainty

Based on six continuous year–round direct measurements of lake E and energy exchange flux, we
determined the E loss during the IC and calibrated and verified different models for E simulation during
the IF and IC. Due to the lack of accurate measurements of deep lake temperature, energy budget closure
ratios of EC observations in QHL are not given in this study. EC measurements have been widely used
to quantify the E of several global lakes, including Lake Superior in America, Great Slave lake in Canada,
Lake Geneva in Switzerland, Lake Valkea−Kotinen in Finland, and Taihu Lake, Erhai Lake, Poyang Lake,
Nam Co, Selin Co and Ngoring Lake in China (Blanken et al., 2000; Vercauteren et al., 2009; Blanken
et al., 2011; Nordbo et al., 2011; Wang et al., 2014; Li et al., 2015; Liu et al., 2015; Guo et al., 2016; Li
et al., 2016; Ma et al., 2016; Lensky et al., 2018). With the most of the known energy budget closure
ratios is over 0.7, EC observation of lakes is regarded as an accurate and reliable direct measurement
method of E, even in lakes over QTP (Wang et al., 2020). Certainly, compared with land stations, the
energy budget closure ratios over lake surfaces can be significantly influenced by the large amount of
heat storage (release) during different seasons (Wang et al., 2020), which would increase the uncertainty



about the quantification of E. Besides, quantification of E during the IC depends on accurate ice
phenology identification, and a longer IC suggests more E. Therefore, the different data sources and
phenological classification methods of ice phenology comprise one source of uncertainty. In addition,
we examined the sensitivity of the input variables (Ta, Ts, Rs and WS) of the chosen model. Increases in
Ta, Ts, Rs, and WS of 10% could result in changes of −2.25%, 1.78%, 10.00% and 10.00% in the
simulated E during the IC, respectively, indicating that E is more sensitive to Rs and WS than Ta and Ts
in the JH model during the IC (Fig. S9). Moreover, the simulated E is minimally sensitive to Ta, and a
10% increase in WS could result in a change of 8.54% in the simulated E during the IF, while a change
in Ts could lead to an exponential change in the simulated E (Fig. S9).
**5. Conclusions**
In summary, based on six continuous year–round 30–min direct flux measurements throughout the cycle
year from 2014 to 2018, the night E of QHL occupied over 40% during both the IF and IC. With a
multiyear average of 175.22 ± 45.98 mm/yr, E during the IC accounted for 23% of the total cycle year E
sum, which is an important component in calculating E of saline lakes. A difference–based control factor
of E was also found during the IF and IC. E of QHL was mainly controlled by atmospheric dynamic
factors (WS, Δe, and P) during the IF, whereas it was driven by both energy exchange and atmospheric
boundary conditions (Rn, Ta−Ts and WS) during the IC. Thus, the MT model based on molecular
diffusion performed best in lake E simulation during the IF, while the JH model based on energy exchange
performed best during the IC. Furthermore, simulation of the E of QHL showed a slight decrease from
2003 to 2017, caused by a decrease in E during the IC, and WS weakening may have resulted in an
average reduction of 11.1% in lake E during the IC from 2003 to 2017. Our results suggest that E during
the IC is non–negligible for saline lake E, and E simulation should be further improved in future model
simulation studies, considering the difference in its potential mechanisms during the IC.
**Author Contributions**
XY Li conceived the idea, FZ Shi performed the analyses. XY Li, FZ Shi, DL Chen and YJ Ma led the
manuscript writing. SJ Zhao, YJ Ma, JQ Wei and QW Liao provided analysis of datasets. All contributed
to review and revise the manuscript.





**Acknowledgements**
The study was financially supported by the National Natural Science Foundation of China
(NSFC: 41971029), the Second Tibetan Plateau Scientific Expedition and Research Program
(STEP, grant no. 2019QZKK0306), the State Key Laboratory of Earth Surface Processes and
Resource Ecology (2021–ZD–03) and Ten Thousand Talent Program for leading young
scientists and the China Scholarship Council. The gridded climate datasets from the interim
reanalysis dataset v5 (ERA5) produced by the European Centre for Medium–Range Weather
Forecasts (https://cds.climate.copernicus.eu/cdsapp#!/search?type=dataset) and the China
Regional High–Temporal–Resolution Surface Meteorological Elements–Driven Dataset
(CMFD) (http://data.tpdc.ac.cn/en/) can be freely accessed. The daily lake ice coverage data
were retrieved from the National Tibetan Plateau Data Center (http://data.tpdc.ac.cn/en/).
**Competing interests**
The contact author has declared that none of the authors has any competing interests.



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





**Figure Legends**

**Figure 1. Location of Qinghai Lake (below) and the measurement site of the Chinese Torpedo Qinghai Lake test base (upper).** The insets in the upper picture are photos of the four–way radiometer and infrared thermometer (left), meteorological variable measurements (middle), and eddy covariance sensors (right).

**Figure 2. Diurnal characteristics of evaporation (E), latent heat flux (LE), sensible heat flux (H), heat storage change (G) and net radiation (Rn) of Qinghai Lake (QHL) during the ice–free (IF) and ice–covered (IC) periods from 2014 to 2018.** The multiday average 30–min data during the IF and IC in each cycle year are shown here, and the colored shading indicates a 0.5 standard deviation. The gray area indicates nighttime. The labels 2014/2015, 2015/2016, 2016/2017, 2017/2018 and 2018/2019 indicate the cycle year of the freeze–thaw cycles.

**Figure 3. Evaporation (E) rate (a, c, and e) and annual E sum (b, d and f) of Qinghai Lake (QHL) during the cycle year (annual: AN), ice–free (IF) and ice–covered (IC) periods in each cycle year from 2014 to 2018.** a and b show daily data, c and d show daytime data, and e and f show nighttime data. The whiskers in a, c and e show the 1.5 interquartile range, while the letter associated with the whiskers indicates statistically significant differences via one–way ANOVA during the different freeze–thaw periods in each year from 2014 to 2018. The labels 2014/2015, 2015/2016, 2016/2017, 2017/2018, and 2018/2019 indicate the cycle year of freeze–thaw cycling.

**Figure 4. Sensitivity coefficient between the daytime and nighttime climatic factors and evaporation (E) rate of Qinghai Lake (QHL) during the ice–free (IF) and ice–covered (IC) periods.** *, ** and *** indicate statistical significance at the $P < 0.1$, $P < 0.05$ and $P < 0.01$ levels, respectively, via Student's t tests. Rn, $\Delta$e. WS, WD, Pres, Ta−Ts, Tl and ICR indicate the net radiation, vapor pressure difference, wind speed, wind direction, Pres, the difference between the air and lake surface temperatures, the average temperature of the lake body from 0 to 300 cm, and ice coverage rate, respectively.

**Figure 5. Interannual variability in the simulated evaporation (E) rate (a−c) and annual E sum (d−f) of Qinghai Lake (QHL) in the cycle year (annual: AN), ice–free (IF), and ice–covered (IC) periods from 2003 to 2017.** The blue shading indicates a 0.5 standard deviation, and the red shading





indicates the 95% confidence interval of the trend line.
**Figure 6. Evaporation (E) and heat storage change (G) in Qinghai Lake (QHL) during the ice–free**
**(IF) and ice–covered (IC) periods.** WS, Pres, $\Delta e$, $Ta-Ts$, Rn, and ICR are the wind speed, air pressure,
vapor pressure difference, difference between $Ta$ and $Ts$, net radiation, and ice coverage rate of the lake,
respectively. The red plus sign indicates a positive effect of the variable on E.
**Figure 7. The multiyear average contribution of the changes in air temperature (Ta), lake surface**
**temperature (Ts), downward shortwave radiation (Rs), and wind speed (WS) to the simulated**
**evaporation (E) of Qinghai Lake (QHL) in the cycle year (annual: AN), ice–free (IF) and ice–**
**covered (IC) periods from 2003 to 2017**. a shows the multiyear average change in the E rate caused by
Ta, Ts, Rs, and WS; b shows the multiyear average change in the annual E sum caused by Ta, Ts, Rs, and
WS; and c shows the multiyear average change percentage of E caused by Ta, Ts, Rs, and WS. The
whiskers indicate a 0.5 standard deviation.





**Figures**
**Figure 1.**

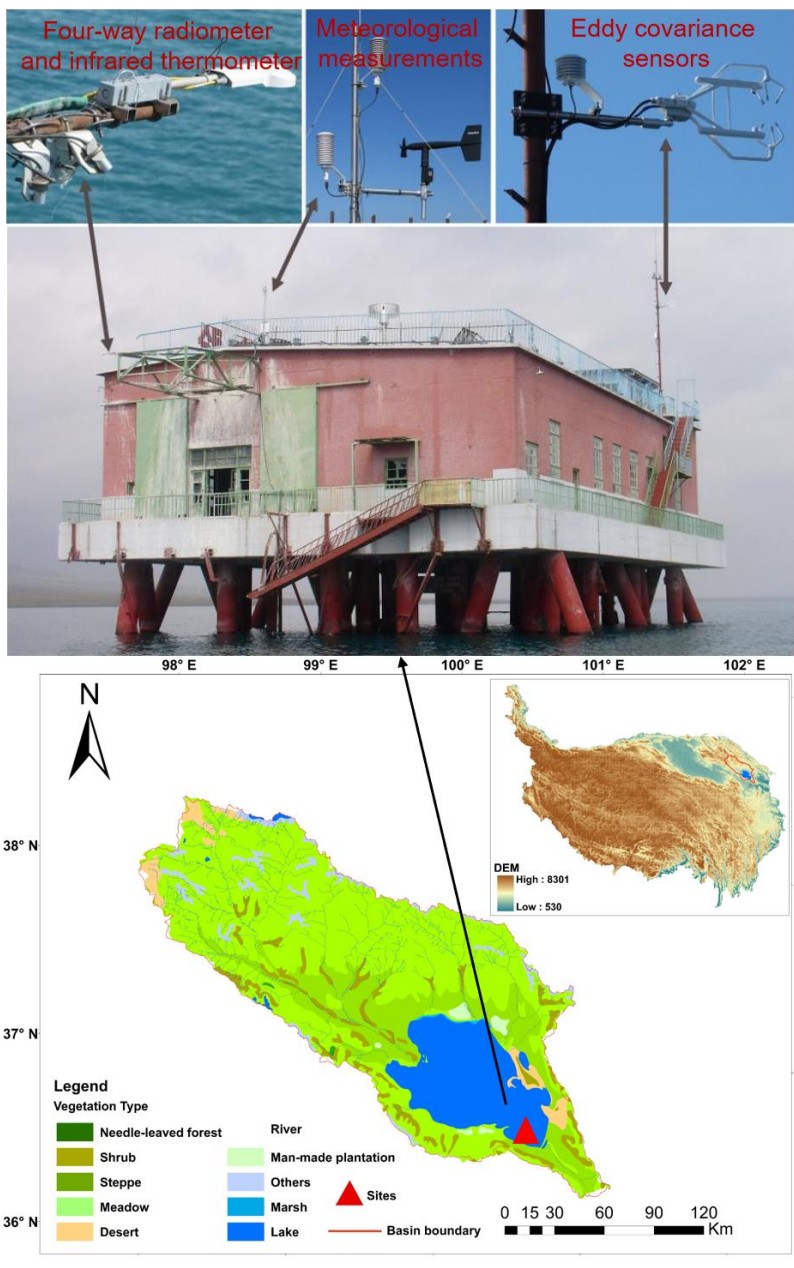






**Figure 2.**

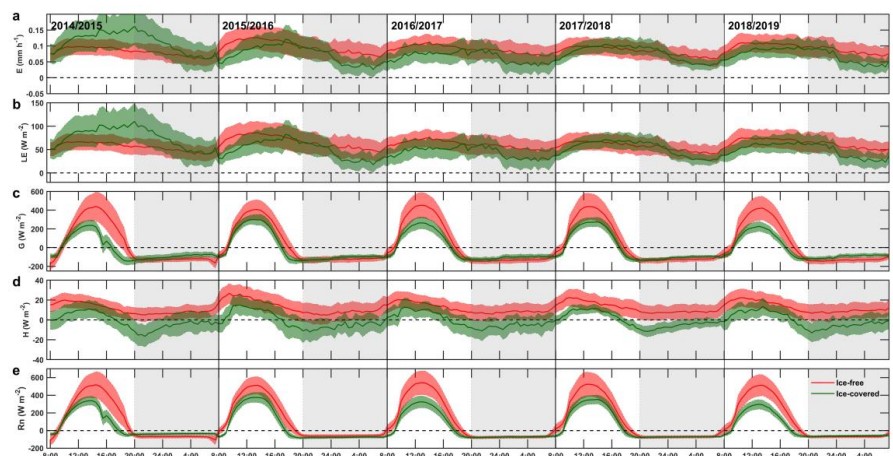








**Figure 3.**

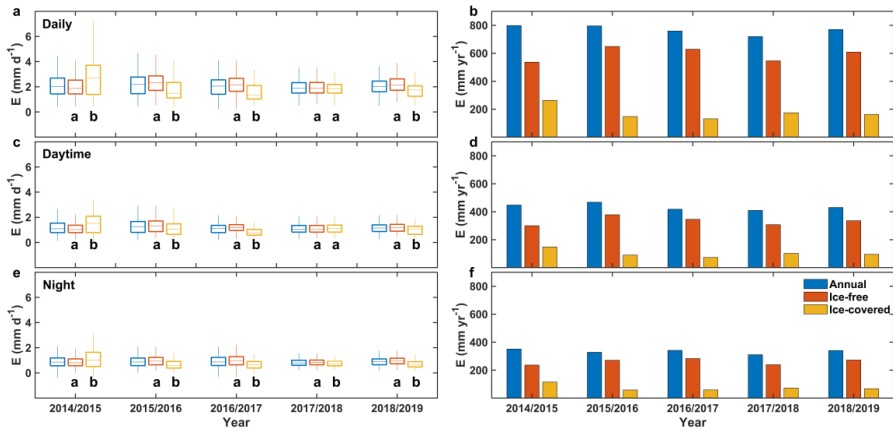







**Figure 4.**

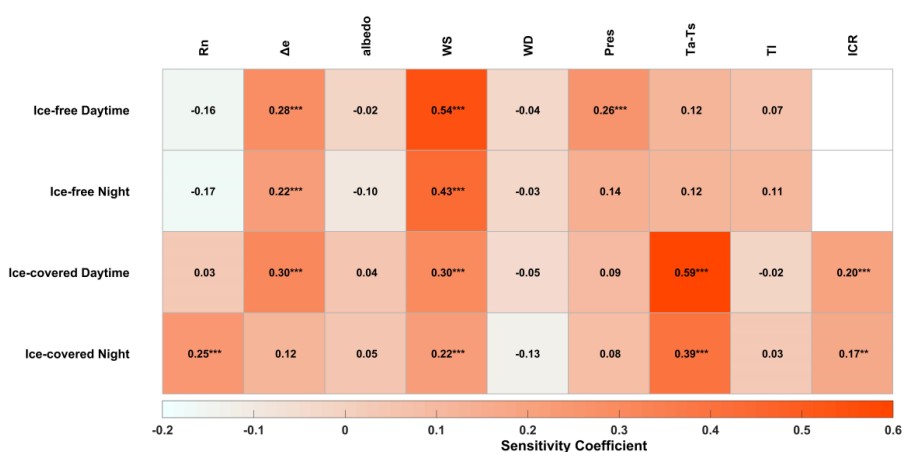







**Figure 5.**

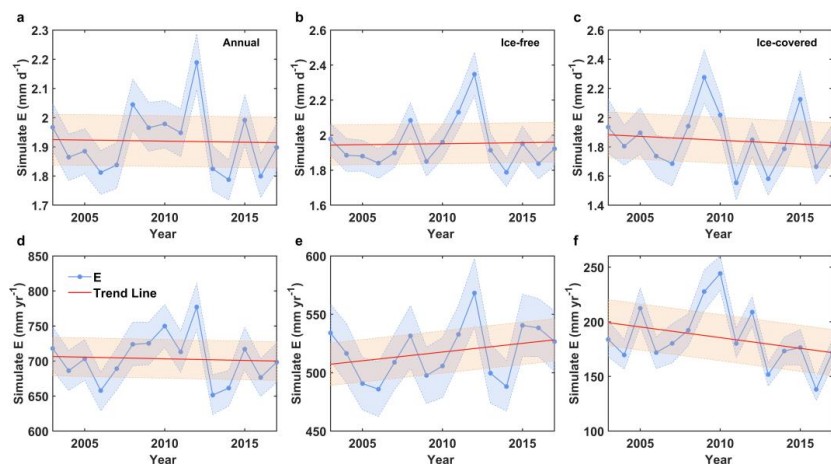





**Figure 6.**

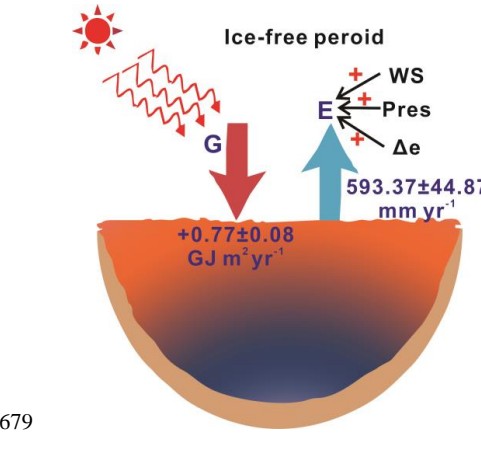
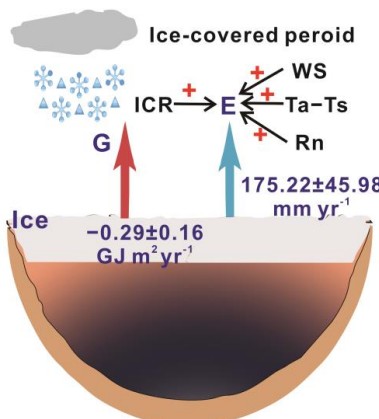





**Figure 7.**

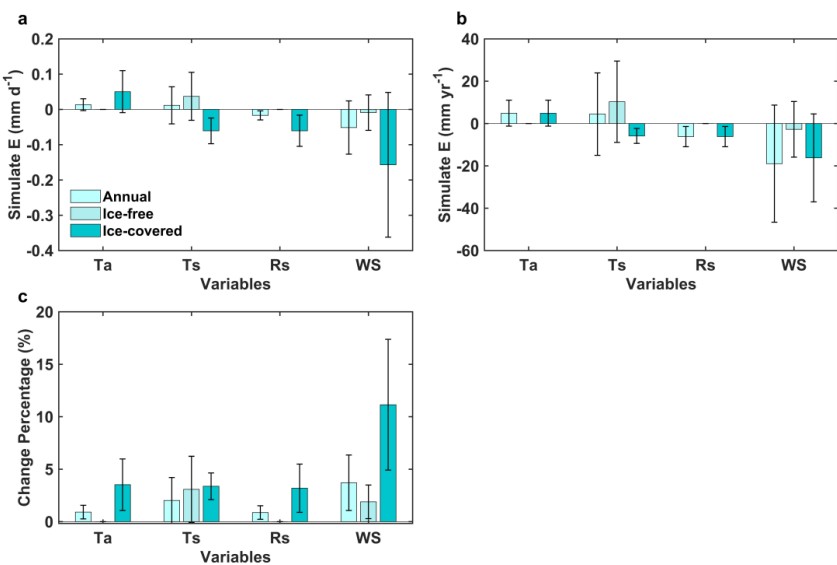

