# Peer review of "Evaporation and sublimation measurement and modelling of an alpine saline lake influenced by"

_Hydrology and Earth System Sciences, 2023_

## Referee Comment (RC1)

The authors quantified evaporation/sublimation (E) during ice-free (IF) and ice-cover (IC) periods for a large lake on the Tibetan Plateau. Field observations were collected between 2014 to 2019 and used to quantify evaporation/sublimation (E) and determine the main controls on E during the IF and IC period and annually. These results were then used to validate and assess three different types of E models (Mass Transfer, atmosphere dynamics and statistical model) to determine which model(s) would be adequate for simulating E during IF, IC and Annual (AN) conditions. The models were introduced to simulate E for the 2003 to 2017 period using reanalysis data to study climate change during IF, IC and annual lake conditions. This paper presents an interesting and innovative contribution to lake E by using 6 years of continuous high-resolution and precious observation datasets. There are not too much papers assessing evaporation from the Tibetan Plateau region or studying sublimation during the ice-covered period. The significance of the results is thus important for improving our understanding of the main controls of E during both IC and IF conditions on an alpine saline lake, and these results can be helpful to improve current hydrological models of alpine lakes. Thus, I recommend this paper for publication in HESS after a major revision. Besides, I did have some concerns about this paper as follows:

Major comments:
(1) The objectives contradict some of the methods. In the second objective, the authors state that two models will be calibrated and verified, however, within the methods section three models are calibrated and verified and not just two models.

(2) Use summary tables for the observed data collection, Reanalysis of datasets, models, and variables. This will make it easier to understand the data collection, cleaning, and processing. Currently, the way these variables and their measurements are presented makes it unclear. For example, in Line 138 it is not clear if the gas analyzer is at the same height as the 3-D sonic anemometer. Besides, the observed meteorological data is in a 30 min timestep; but ERA-5 is in a 1-hr timestep. How was this addressed when assessing the fit between the observed data and the reanalysis data?

(3) E values for Antarctica are in mm/month during IC, Lines 346-347 you present the annual sum of E; but to draw comparisons to Antarctica can you put this value into monthly for the IC period? The total value does show it is larger but by showing it in the same units as Antarctica it will be easier to see how it relates monthly

(4) In the key findings you state that wind weakening is considered a key finding; however, wind weakening and its relationship to E during the IC period is not discussed. As this is considered a key finding this should be discussed.

Minor comments:
(1) Line 37: did the result for IC consider ice loss?
(2) Line 132: you should reference your site in Figure 1.
(3) Line 166: Long time should be long-time.

(4) Lines 178-183: Qui et al 2019 is the referenced method for the ice phenology dataset, however, how do they account for the accuracy of the ice dataset you are using for your analysis? Using visible MODIS to ascertain freeze dates can be difficult, as the ice must be substantial enough to change the reflective properties. A few brief sentences to expand on the methods in this section would do well to provide context for the accuracy of the ice dataset you are using.

(5) Fig S3: the x-axis should be the same for all 3 figures. They should all range from 0 to 60%; if you are to just glance at the figures and not read the axis label/units one would assume they all contribute the same during each period.

(6) Fig S5: the y-axis should have the same scale for all figures. Why is the x-axis for ice cover 1 year? Whereas the IF and AN showing 3 and 4 years respectively? Your caption states they are showing the results from 2014-2018.

(7) Fig 1: DEM needs units, missing the line for rivers in the legend, is the scale the same for the inset map?

(8) When using the abbreviations for ice-covered (IC) or ice-free (IF), they are missing context (or a word) such as conditions or periods.

---

## Author Response (AR1)

Dear Editor,

We appreciate very much the valuable and constructive comments on our manuscript entitled **"Evaporation and sublimation measurement and modelling of an alpine saline lake influenced by freeze–thaw on the Qinghai–Tibet Plateau" (ID hess-2023-100)** from you and two reviewers. We addressed each comment very carefully and also provide for each comment what changes have been done in the revised manuscript or a reasonable explanation to your and the reviewer's observation. The following paragraphs respond to the specific comments of referee, the original review comments are listed first in their originals (in italic), followed by our itemized responses (in bold). **The line number mentioned in the responses is based on the manuscripts without tracks.**

We hope that the editor agrees that the changes we have done fully address the reviewer's comments and that the manuscript can therefore be accepted as it is.

Sincerely yours,

Authors of the Manuscript

**Point to Point Response to the Editor's Comments**

**Major Comments:**

*1. I still do not understand where salinity is taken as a parameter in this study. I see only one parameter (alpha, L229) which leads to a reduction of 3% regarding the saturated vapor pressure. This paper is motivated by the fact that there are only few studies dealing with evaporation/sublimation in saline lakes but I do not see where salinity is investigated. More problematic: why do you think that the salinity is playing a role on the sublimation during the ice-covered period? The sublimation is there controlled by the atmosphere and ice parameters (not by the water).*

**Response: Thank you for your elaborate comments. Yes, we considered the effect of salinity on evaporation during the ice-free periods (IFP). And it was applied to the MT and AD model for E simulation during IFP, but not considered during ice-covered periods (ICP).**

**As mentioned in this study, evaporation is mainly controlled by the wind speed, vapor pressure difference, and air pressure during IFP, but driven by the net radiation, the difference between the air and lake surface temperatures, wind speed, and ice coverage during the ICP in Qinghai Lake (QHL). And salinity mainly reduces the free energy of water molecules and results in a reduced saturated vapor pressure above saline lakes, which is closely related to the water vapor pressure difference. Thus, salinity mostly affects evaporation during IFP in QHL.**

**In addition, although the application of water activity has little effect on the evaporation value of QHH (a decrease of 3% and approximate 24 mm/yr), we think that it is important to consider the effect of salinity on the evaporation simulation for the explanation of the mechanism in models of saline lakes. Because this influence increases gradually over saline. When the salinity concentrations are 100 g/L and 300 g/L, the reductions in evaporation are 3.4% and 31.9%, respectively. Thus, it is more reasonable to consider the effect of salinity on evaporation in saline lakes.**

Following your suggestion, we clarified our expression in the section of '2.6. Models for daily lake evaporation simulation' (L236~248) as following: **Considering that Qinghai Lake is a saline lake, and many studies have pointed out that it is valuable to consider the influence of salinity on saline lake evaporation, and with the increase of salinity, it will exert greater inhibition on evaporation (Hamdani et al., 2018; Mor et al., 2018). Thus, the water activity coefficient ($\alpha$) which is defined as the ratio between the vapor pressure above saline water and that above freshwater at the same temperature has been introduced to characterize the effect of salinity on saline lake evaporation (Salhotra et al., 1987; Lensky et al., 2018). Because saline water drains out salt during freezing (Badawy, 2016), we only introduced the $\alpha$ into the evaporation simulation of Qinghai Lake during IFP.**

*2. Similarly, I do not understand how Ts (surface temperature) is estimated. Ts should be the lake water temperature at the surface during the ice-free period and the ice surface temperature during the ice-covered period. This is not clear in your manuscript.*

**Response: Thank you for your elaborate comments. Two kinds of Ts were used in this study: the observed Ts measured with an infrared thermometer (model SI–111, Campbell Scientific Inc.) (L158~159) and the Ts acquired from ERA5 (L170~173).**

**The observed Ts has a temporal resolution of 30–min and covered the period of January 1, 2014 to December 31, 2019. And we calculated the daily Ts by averaging the 30–min observed data for each day, the daytime (nighttime) energy exchange flux and climate data were calculated by averaging the 30–min observed data of 8:00 am to 7:30 pm (8:00 pm to 7:30 am), which was used to explore the key factor controlling lake E during the daytime and nighttime of IFP and ICP (L280~288). To make it clear, we modified the expression about the statistical analysis of observed data, which was shown as: Second, we calculated the multiday average 30–min energy exchange flux during the IFP and ICP in each year to evaluate the basic statistical characteristics of the diurnal E and exchange flux. The daily energy exchange flux and climate data were calculated by averaging the 30–min**

**observed data** **for each day, the daytime (nighttime) energy exchange flux and climate data were calculated by averaging the 30–min** **observed data** **of 8:00 am to 7:30 pm (8:00 pm to 7:30 am).**

The ERA5 Ts was actually the skin temperature product in ERA5, which has a spatial resolution of 0.1° and a temporal resolution of hourly, and covered the period of January 1, 2001 to December 31, 2018. We extracted the ERA5 skin temperature pertaining to QHL via a grid mask with a spatial resolution of 0.1° and averaged the data in all pixels (L172~174). And we adjusted the ERA5 Ts by comparing it and the observed Ts at daily scale ($R^2 = 0.92$, P < 0.01) (Fig S1; equation (1)). Because the period (six years) of observed datasets is too short, the ERA5 Ts was used to drive the lake E models, and to analyze the interannual variation of evaporation and its response to climate variability.

To make the information of those datasets clear, following the suggestion of you and reviewer#1, we added a summary table (Table S1 in this revision) which contains the instrument type, height from the lake surface or spatial resolution, time resolution and purpose of each variable from observed, reanalysis, model and remote sensing datasets.

Table S1. The information about variables from observed, reanalysis, model and remote sensing datasets.

| Dataset | Instrument type | Height from the lake surface/Spatial resolution | Time resolution | Purpose |
|---|---|---|---|---|
| Observed H and LE | EC system (Three–dimensional sonic anemometer: CSAT3, Campbell, USA, and open–path infrared gas analyzer: EC150, Campbell, USA) | 17.3 m | 30 min | Evaporation and energy calculation, and model calibration and verification |
| Observed Ta, RH and Pres | HMP155, Vaisala, Finland | 12.5 m | 30 min | Analysis of |
| Observed WS and WD | 05103, R.M. Young, USA | 12.5 m | 30 min | evaporation |
| Observed Ts | SI−111, Campbell, USA | 0 | 30 min | influence factors |

| Observed Tl | 109L, Campbell, USA | −0.2 to −3.0 m | 30 min | |
| Observed precipitation | TE525, Campbell, USA | 10 m | 30 min | |
| Observed four-component radiometer | CNR4, Kipp&Zonen, Netherlands | 10 m | 30 min | |
| ERA5 Ts | \ | 0.1° | hourly | |
| ERA5 WS | \ | 0.1° | daily | Model input |
| CMFD Ta, Pres, RH and Rs | \ | 0.1° | daily | |
| Lake ice coverage | \ | \ | daily | Lake ice phenology dividing |

Notes: H, LE, Ta, RH, Pres, WS, WD, Ts, Tl and Rs are the abbreviation of sensible heat, latent heat, air temperature, relative humidity, air pressure, wind speed, wind direction, lake surface temperature, water temperature and downward shortwave radiation, respectively. ERA5 and CMFD mean the interim reanalysis dataset v5 and China Regional High–Temporal–Resolution Surface Meteorological Elements–Driven Dataset, respectively. Four-component radiometer is the incoming shortwave, reflected shortwave, and incoming and outgoing longwave radiation.

*3. I also do not understand why you are using ERA5 and/or CMFD model output for the atmospheric forcing. You have a fantastic in-situ facility so why using atmospheric model that will lead to major uncertainties into the fluxes. I also do not see any comparison between the atmospheric model output and your observations (air temperature, radiation, wind speed and other meteorological parameters).*

**Response: Thank you very much for your constructive comments. The period of observed datasets is from 2014 to 2019 (six years), which is too short to analyze the interannual variation of evaporation and its response to climate variability. Thus, we used ERA5 and/or CMFD model output for the atmospheric forcing.**

**The ERA5 and CMFD datasets developed based on land station data have been recognized as the best currently available reanalysis products and have been widely applied in land–surface and hydrological modelling studies in China (Ma et al., 2016; Zhu et al., 2016; Tian et al., 2021; Xiao and Cui, 2021). Comparing**

and adjusting the datasets form ERA5 and/or CMFD model output by the observed data can validate the availability (or accuracy) of the datasets form ERA5 and/or CMFD model output and reduce uncertainty caused by data input in lake E simulation. Thus, following your suggestion, the daily lake surface temperature and WS from EAR5, Ta Rs, RH and Pres from CMFD for QHL were adjusted with fitting equations of the observed daily Ts ($R^2$ = 0.92, P < 0.01), WS ($R^2$ = 0.55, P < 0.01), Ta ($R^2$ = 0.90, P < 0.01), Rs ($R^2$ = 0.73, P < 0.01), RH ($R^2$ = 0.63, P < 0.01) and Pres ($R^2$ = 0.95, P < 0.01) from 2014 to 2018 (Fig. S1) (L177~190).

[Figure]

Fig. S1. Relationship between the daily air temperature (Ta) from the China Regional High–Temporal–Resolution Surface Meteorological Elements–Driven Dataset (CMFD), lake surface temperature (Ts) from the interim reanalysis dataset v5 (ERA5), downward shortwave radiation (Rs) from CMFD, daily wind speed (WS) from ERA5, daily relative humidity (RH) and air pressure (Pres) from CMFD with the observed Ta (a), Ts (b), Rs (c), WS (d), RH (e), and Pres (f) in Qinghai Lake.

*4. In the end, this study provides an interesting sensitivity analysis of the parameters already known as controlling the evaporation/sublimation under specific conditions (the Qinghai-Tibet Plateau). Finally, the writing could be improved.*

**Response: Thank you for your elaborate comments. It is not reasonable to conduct sensitivity analysis of parameters when the control factors are known. Thus, we deleted it in this revision, and reorganized the discussion of the limitation of this study (L463~473).**

**Furthermore, to improve the writing of this manuscript, we edited it for proper**

**English language, grammar, punctuation, spelling, and overall style by one or more of the highly qualified native English speaking editors at Springer Nature Author Services (Figure below).**

[Figure]

FigR1 The certifies of the polish of this manuscript

**Minor comments**

*1. Title. Maybe indicate evoration and submimation*

**Response: Many thanks for your good suggestion, we changed the Title to be 'Evaporation and sublimation measurement and modelling of an alpine saline lake influenced by freeze–thaw on the Qinghai–Tibet Plateau' in this revision.**

*2. L30 "profoundly" is perhaps an overstatement*

**Response: Many thanks for your good suggestion, we deleted this word in this revision.**

*3. I found confusing to merge Evaporation and Sublimation. Maybe distinguish more clearly between Evaporation, E and Sublimation, S.*

**Response: Thank you very much for your elaborate suggestion. We have carefully thought about this problem for a long time. It is true that evaporation mostly refers**

to the phase transition of water from liquid to gas. Sublimation is the phase transition of water from solid to gas, which maybe more suitable for ICP. However, if we used evaporation (E) in IFP and sublimation (S) in ICP, it is hard to deal with the year-round evaporation. Consecrating that lake evaporation can be defined as the exchange of water vapor and energy at the interface between open water and air, many studies also used 'evaposublimation' or 'evapo-sublimation' for the sublimation during ICP (Froyland, 2013; Herrero and Polo, 2016; Guo et al., 2021). That is, sublimation can be regarded as a kind of evaporation. Thus, we defined E as evaporation under ice–free and sublimation under ice–covered conditions at the begging of the abstract (L30).

*4. L38-41. But the controlling parameters are already known. What about the other parameters indicated on L60?*

**Response: Thank you for your insightful comments. As we all know, evaporation is energy-driven, and the diffusive nature also lends itself to mass transfer (or "bulk aerodynamic") formulations. Thus, the variables related to the energy balance and mass transport are the potential influencing factors of lake evaporation. Lake depth, temperature, stratification, thermal stability, and hydrodynamics affect the transmission and storage of energy inside the lake, alter the lake surface temperature and energy balance, and then indirectly affect the evaporation of the lake. Similarly, they also affect the vertical exchange of salinity in saline lake, which in turn regulate lake surface salinity, and then indirectly affect the evaporation of the lake.**

*5. L41 (and discussion) why only looking at wind reduction?*

**Response: Many thanks for your elaborate comments. In addition to wind speed (WS), we also discuss the effects of other climatic variability (such as Ts and solar radiation) on evaporation in Qinghai Lake (L417~424), but highlight the decrease of E caused by wind reduction.**

**In this study, we found that, from 2003 to 2017, E decreased at an average rate of −6.48 ± 4.77 mm/yr (3.23%) and −11.17 ± 14.29 mm/yr (7.56%) due to the decrease in Rs and WS during the ICP, respectively (Fig. 7; Table S3); while the**

**increase in Ts increased E at an average rate of 13.58 ± 20.75 mm/yr (3.54%) during the IFP (Fig. 7; Table S3), which mean WS had the greatest contribution (a relative contribution was 61% of E decrease, comparing to Rs, Ta and Ts) on the decrease of E during the ICP (Fig. 7; Table S3).**

**Besides, numbers of studies show that the global warming has led to an increase in global lake evaporation (Wang et al., 2018; Woolway et al., 2020). The seasonal asymmetry of lake evaporation change affected by different climate and environmental factors may slow down the increase in E caused by global warming. Such as, the decrease in E caused by WS reduction during ICP. I think this is a vital issue at the background of a continued global warming in future study. Thus, we chose to highlight the decrease of E caused by wind reduction.**

[Figure]

Figure 7. The multiyear average contribution of the changes in air temperature (Ta), lake surface temperature (Ts), downward shortwave radiation (Rs), and wind speed (WS) to the simulated evaporation (E) of Qinghai Lake (QHL) in the cycle year (annual: AN), ice–free and ice–covered periods (IFP and ICP) from 2003 to 2017. a shows the multiyear average change in the E rate caused by Ta, Ts, Rs, and WS; b shows the multiyear average change in the annual E sum caused by Ta, Ts, Rs, and WS; and c shows the multiyear average change percentage of E caused by Ta, Ts, Rs, and WS. The whiskers indicate a 0.5 standard deviation.

*6. L76 what do you mean by growing seasons?*

**Response: Thank you for your insightful comments. The growing seasons mean the period of vegetation growth which covers mid–May to mid–October at mid-high latitudes in the Northern Hemisphere. Yes, it is not appropriate in this study, so we replaced the 'growing seasons' by 'ice-free period' (L77).**

*7. L151 water temperature at 0.2 m, I guess this sensor and the other underneath are frozen in the ice in winter?*

**Response: Thank you for your constructive comments. The water temperature was measured with five temperature probes (109 L, Campbell Scientific Inc.) at depths of 0.2, 0.5, 1.0, 2.0 and 3.0 m. The five temperature probes were fastened to a heavy steel rope that leads to the bottom of the lake. Since the maximum ice depth at the observation site is approximate 80 cm, the two probes for lake temperature at depths of 0.2 and 0.5 cm were frozen in the ice dur ice-covered periods. The ambient temperature of those probes is -50~100 °C, and the range is -40~70 °C, so the freeze of probe does not affect the observation accuracy.**

*8. P10. Are those models the one mostly used in limnology? Maybe you could review the models used in limnology. See for instance Xie et al. 2023 for the ice-covered period, https://doi.org/10.1016/j.jhydrol.2023.129461)*

**Response: Thank you for your elaborate comments. Yes,the mass–transfer model (MT model), atmospheric dynamics model (AD model) and empirical formula based on statistical analysis (such as Jensen–Haise method: JH model) are traditional and basic methods for lake evaporation simulation, which have been widely used in lake evaporation estimation over the world (Dalton, 1802; Sartori, 2000; Gianniou and Antonopoulos, 2007; Pillco et al., 2019; Liu, 2023). Indeed, the 1D lake thermodynamics model also has been used for the simulation of lake ice thickness and energy balance (ice sublimation) in ice-covered periods (Pour et al., 2017; Stepanenko et al., Xie et al., 2023). However, in this study we were interested in verifying the consistency of the accuracy of the traditional models for the evaporation simulation during ice-free periods and ice-covered periods, because**

almost all models were calibrated and verified against evaporation observations during the ice-free periods, while evaporation (or sublimation) during the ice-covered periods was either not calculated or unverified. Thus, there traditional and representative were chosen and used in this study.

Considering that the 1D lake thermodynamics model can quantify the energy transfer and balance at the vertical depth of the lake, we are designing the observation system of lake thermodynamics parameters, such as sampling of lake ice in ice-covered periods (Figure below), and verify and develop a suitable 1D or even 3D lake thermodynamics evaporation models for Qinghai Lake (or even lakes in the Qinghai-Tibet Plateau) in the next study. And following your comments, we pointed out that ignoring the 1D lake thermodynamics model for ice sublimation was indeed one of the limitations of this study (L465~473).

[Figure]

FigR2 Sampling and measurement of lake ice of Qinghai Lake in Feb 2023.

*9. L268 "severally"?*

**Response: Done.**

*10. L330 - L338 What is the point of the text here? I suggest to delete*

**Response: Thank you very much for your suggestion. In fact, we try to explain why there is a similar rate of evaporation in ice-free periods and sublimation in ice-covered periods by microcosmic diffusion mechanism of liquid droplets and ice crystals. It's true that it is too hard to make it clear. Thus, following your suggestion, we have deleted this part in our revision.**

**References**

Badawy, S. M. (2016). Laboratory freezing desalination of seawater. Desalination and Water Treatment, 57(24), 11040-11047.

Dalton, J. (1802). Experimental essays on the constitution of mixed gases; on the force of stream or vapor from water and other liquids, both in a Torricellian vacuum and in air; on evaporation; and on the expansion of gases by heat. Proceedings of Manchester Literary and Philosophica Society, 5, 536–602.

Froyland, H. K. (2013). Snow loss on the San Francisco peaks: Effects of an elevation gradient on evapo-sublimation (Doctoral dissertation, Northern Arizona University).

Gianniou, S. K., & Antonopoulos, V. Z. (2007). Evaporation and energy budget in Lake Vegoritis, Greece. Journal of Hydrology, 345(3-4), 212-223.

Guo, S., Chen, R., Han, C., Liu, J., Wang, X., & Liu, G. (2021). Five-Year Analysis of Evaposublimation Characteristics and Its Role on Surface Energy Balance SEB on a Midlatitude Continental Glacier. Earth and Space Science, 8(12), e2021EA001901.

Hamdani, I., Assouline, S., Tanny, J., Lensky, I. M., Gertman, I., Mor, Z., & Lensky, N. G. (2018). Seasonal and diurnal evaporation from a deep hypersaline lake: The Dead Sea as a case study. Journal of Hydrology, 562, 155–167.

Herrero, J., & Polo, M. J. (2016). Evaposublimation from the snow in the Mediterranean mountains of Sierra Nevada (Spain). The Cryosphere, 10(6), 2981-2998.

Koutsoyiannis, D. (2020). Revisiting the global hydrological cycle: is it intensifying?. Hydrology and Earth System Sciences, 24(8), 3899-3932.

Lensky, N. G., Lensky, I. M., Peretz, A., Gertman, I., Tanny, J., & Assouline, S. (2018). Diurnal Course of evaporation from the dead sea in summer: A distinct double peak induced by solar radiation and night sea breeze. Water Resources Research, 54(1), 150–160.

Liu, Z. (2023). Accuracy of methods for simulating daily water surface evaporation evaluated by the eddy covariance measurement at boreal flux sites. Journal of Hydrology, 616, 128776.

Ma, N., Szilagyi, J., Niu, G. Y., Zhang, Y., Zhang, T., Wang, B., & Wu, Y. (2016). Evaporation variability of Nam Co Lake in the Tibetan Plateau and its role in recent rapid lake expansion. Journal of Hydrology, 537, 27–35.

Mor, Z., Assouline, S., Tanny, J., Lensky, I. M., & Lensky, N. G. (2018). Effect of water surface

salinity on evaporation: The case of a diluted buoyant plume over the Dead Sea. Water Resources Research, 54(3), 1460–1475.

Pillco Zolá, R., Bengtsson, L., Berndtsson, R., Martí-Cardona, B., Satgé, F., Timouk, F., ... & Pasapera, J. (2019). Modelling Lake Titicaca's daily and monthly evaporation. Hydrology and Earth System Sciences, 23(2), 657-668.

Salhotra, A. M., Adams, E. E., & Harleman, D. R. (1987). The alpha, beta, gamma of evaporation from saline water bodies. Water Resources Research, 23(9), 1769–1774.

Sartori, E. (2000). A critical review on equations employed for the calculation of the evaporation rate from free water surfaces. Solar energy, 68(1), 77-89.

Tian, W., Liu, X., Wang, K., Bai, P., & Liu, C. (2021). Estimation of reservoir evaporation losses for China. Journal of Hydrology, 596, 126142.

Wang, W., Lee, X., Xiao, W., Liu, S., Schultz, N., Wang, Y., ... & Zhao, L. (2018). Global lake evaporation accelerated by changes in surface energy allocation in a warmer climate. Nature Geoscience, 11(6), 410-414.

Woolway, R. I., Kraemer, B. M., Lenters, J. D., Merchant, C. J., O'Reilly, C. M., & Sharma, S. (2020). Global lake responses to climate change. Nature Reviews Earth & Environment, 1(8), 388–403.

Xiao, M., & Cui, Y. (2021). Source of evaporation for the seasonal precipitation in the Pearl River Delta, China. Water Resources Research, e2020WR028564.

Zhu, L., Yang, K., Wang, J., Lei, Y., Chen, Y., Zhu, L., ... & Qin, J. (2016). Quantifying evaporation and its decadal change for Lake Nam Co, central Tibetan Plateau. Journal of Geophysical Research: Atmospheres, 121(13), 7578–7591.

**Point to Point Response to the Reviewer#1's Comments**

*The authors quantified evaporation/sublimation (E) during ice-free (IF) and ice-cover (IC) periods for a large lake on the Tibetan Plateau. Field observations were collected between 2014 to 2019 and used to quantify evaporation/sublimation (E) and determine the main controls on E during the IF and IC period and annually. These results were then used to validate and assess three different types of E models (Mass Transfer, atmosphere dynamics and statistical model) to determine which model(s) would be adequate for simulating E during IF, IC and Annual (AN) conditions. The models were introduced to simulate E for the 2003 to 2017 period using reanalysis data to study climate change during IF, IC and annual lake conditions. This paper presents an interesting and innovative contribution to lake E by using 6 years of continuous high-resolution and precious observation datasets. There are not too much papers assessing evaporation from the Tibetan Plateau region or studying sublimation during the ice-covered period. The significance of the results is thus important for improving our understanding of the main controls of E during both IC and IF conditions on an alpine saline lake, and these results can be helpful to improve current hydrological models of alpine lakes. Thus, I recommend this paper for publication in HESS after a major revision. Besides, I did have some concerns about this paper as follows:*

**Response: Thank you very much for your positive comments on the significance of this study. Your comments do improve our manuscript, and we provide a point-to-point response to your comments in bold font below, and revisions were annotated in the manuscript in underline font.**

**Major comments:**

*1. The objectives contradict some of the methods. In the second objective, the authors state that two models will be calibrated and verified, however, within the methods section three models are calibrated and verified and not just two models.*

**Response: Thank you for your insightful comments. Yes, the inappropriate expression in the second objective led to a misunderstanding which contradicts some of the methods. Actually, as you mentioned above, based on six years of**

observational data, we validated and assessed three different types of E models (Mass Transfer, atmosphere dynamics and statistical model) to determine which model(s) would be adequate for simulating E during ice–free, ice–covered and Annual conditions. And then, we select an optimal model for E simulation in ice–free and ice–covered periods (IFP and ICP) according to the maximum $R^2$ and the minimum RMSE, respectively. The result shows that the mass transfer model simulates lake E well during the IFP, and the model based on energy achieves a good simulation during the ICP.

Thus, we have modified this expression in the second objective shown as follows (L116~119): In addition, combined with reanalysis climate datasets, a mass transfer model (MT model), an atmospheric dynamics model (AD model), and a model based on energy, temperature and WS (JH model) were calibrated and verified, with the optimal model chosen for the simulation of lake E and its response to climatic variability during the IFP and ICP from 2003 to 2017.

*2. Use summary tables for the observed data collection, Reanalysis of datasets, models, and variables. This will make it easier to understand the data collection, cleaning, and processing. Currently, the way these variables and their measurements are presented makes it unclear. For example, in Line 138 it is not clear if the gas analyzer is at the same height as the 3-D sonic anemometer. Besides, the observed meteorological data is in a 30 min timestep; but ERA-5 is in a 1-hr timestep. How was this addressed when assessing the fit between the observed data and the reanalysis data?*

Response: Many thanks for your good suggestions and constructive comments. Following your suggestion, we added a summary table (Table S1 in this revision) which contains the instrument type, height from the lake surface or spatial resolution, time resolution and purpose of each variable from observed, reanalysis, model and remote sensing datasets.

The instruments of the measurement of the energy exchange flux and micrometeorological parameters were installed at the China Torpedo Qinghai Lake test base which has a height of 10 m above the water surface (Fig. 1), so most of the observed variables have a height over 10 m above the water surface the

**concrete height of variables was listed in Table S1.**

**In this study, all analyses are based on daily datasets, except for the analysis of diurnal variation of evaporation and energy by a time resolution of 30 min in section 3.1. Thus, we generated the daily EAR5 Ts by averaging the hourly temperature over 24 h per day. In order to make the methods clearer, we have added the following statement in the methods section of this revision (L299~300): The analysis of partial least squares regression, random forest methods, and E simulation, calibration and verification were conducted at the daily scale.**

Table S1. The information about variables from observed, reanalysis, model and remote sensing datasets.

| Dataset | Instrument type | Height from the lake surface/Spatial resolution | Time resolution | Purpose |
|---|---|---|---|---|
| Observed H and LE | EC system (Three–dimensional sonic anemometer: CSAT3, Campbell, USA, and open–path infrared gas analyzer: EC150, Campbell, USA) | 17.3 m | 30 min | Evaporation and energy calculation, and model calibration and verification |
| Observed Ta, RH and Pres | HMP155, Vaisala, Finland | 12.5 m | 30 min | |
| Observed WS and WD | 05103, R.M. Young, USA | 12.5 m | 30 min | |
| Observed Ts | SI−111, Campbell, USA | 0 | 30 min | Analysis of evaporation influence factors |
| Observed Tl | 109L, Campbell, USA | −0.2 to −3.0 m | 30 min | |
| Observed precipitation | TE525, Campbell, USA | 10 m | 30 min | |
| Observed four-component radiometer | CNR4, Kipp&Zonen, Netherlands | 10 m | 30 min | |
| ERA5 Ts | \ | 0.1° | hourly | |
| ERA5 WS | \ | 0.1° | daily | Model input |
| CMFD Ta, Pres, RH and Rs | \ | 0.1° | daily | |
| Lake ice coverage | \ | \ | daily | Lake ice |

Notes: H, LE, Ta, RH, Pres, WS, WD, Ts, Tl and Rs are the abbreviation of sensible heat, latent heat, air temperature, relative humidity, air pressure, wind speed, wind direction, lake surface temperature, water temperature and downward shortwave radiation, respectively. ERA5 and CMFD mean the interim reanalysis dataset v5 and China Regional High–Temporal–Resolution Surface Meteorological Elements–Driven Dataset, respectively. Four-component radiometer is the incoming shortwave, reflected shortwave, and incoming and outgoing longwave radiation.

*3. E values for Antarctica are in mm/month during IC, Lines 346-347 you present the annual sum of E; but to draw comparisons to Antarctica can you put this value into monthly for the IC period? The total value does show it is larger but by showing it in the same units as Antarctica it will be easier to see how it relates monthly.*

**Response: Many thanks for your good suggestion! Yes, it would be clearer to draw comparisons at the same unit. Due to the ICP varied from 83 to 97 during 2014~ to 2018, we estimated monthly E of ICP by multiplying the mean daily E of ICP by 30, and added the estimated results as additional reference data in this revision (L367~370) shown as follows: In this study, we found that the E sum ranges from 130.59 to 262.45 mm during the ICP (approximately 51.60 to 81.3 mm/month, by multiplying the mean daily E of ICP by 30) from 2014 to 2018, which is higher than the previous observations from Antarctic ice sheets or lakes.**

*4. In the key findings you state that wind weakening is considered a key finding; however, wind weakening and its relationship to E during the IC period is not discussed. As this is considered a key finding this should be discussed.*

**Response: Thank you very much for your constructive comments. We agree with referee at this point. It is very important to do some discussion of wind weakening and its relationship to E during the ICP. As we all know, E of lake driven by energy and is also a process of molecular diffusion which lends itself to mass transfer. Thus, the direct influences on lake E are energy, water vapor pressure difference and air stability above water. And wind stilling would enhance the stability of the atmosphere above the water surface, which in turn inhibits evaporation.**

Following your suggestion, we have reorganized the discussion of the effects of climate variability on E, which described the studies of climate change on the QTP, discussed the effects of changes in wind speed and other climatic factors on E, and compared our results with studies of Selin Co and Namu Co.

We added the discussion in this revision (L417~432) shown as follows: **Furthermore, the QTP has been suffering surface air warming and moistening, solar dimming, and wind stilling since the beginning of the 1980s across the QTP (Yang et al., 2014; Kuang and Jiao, 2016), which affects the hydrothermal processes of the lake, such as increasing Ts and shortening lake ice phenology (Wan et al., 2018; Cai et al., 2019). An increase in Ts enhances the diffusion of water molecules and enlarges $\Delta e$ between the water surface and the air, which in turn promotes evaporation (Wang et al., 2018; Woolway et al., 2020), while a reduction in solar radiation decreases the energy input of the lake, and wind stilling enhances the stability of the atmosphere above the water surface, which in turn inhibits evaporation (Roderick and Farquhar, 2022; Guo et al., 2019). We found a decrease in E during the AN from 2003 to 2017, due to the steeper decrease in E caused by solar dimming and wind stilling during the ICP than the increase engendered by the increase in Ts during the IFP. From 2003 to 2017, E decreased at an average rate of −6.48 ± 4.77 mm/yr (3.23%) and −11.17 ± 14.29 mm/yr (7.56%) due to decrease in Rs and WS during the ICP, respectively (Fig. 7; Table S3), while the increase in Ts increased E at an average rate of 13.58 ± 20.75 mm/yr (3.54%) during the IFP (Fig. 7; Table S3). Previous studies have found similar results in Selin Co and Namu Co (Zhu et al., 2016; Guo et al., 2019). For example, Guo et al. (2019) found that E was mainly controlled by WS, and a decrease in WS led to a decrease in E from 1985 to 2016 in Selin Co.**

**Minor comments:**

*1. Line 37: did the result for IC consider ice loss?*

**Response: Yes, E was observed by an eddy covariance observation system installed at the China Torpedo Qinghai Lake test base, which is based on the principle of**

eddy correlation, and can direct measure the water vapor flux, the latent heat, and the sensible heat of the lake surface in the spatial range of 100~1000 m in real time. Thus, E in this paper includes evaporation under ice–free and sublimation under ice–covered conditions mentioned in Abstract and Introduction section.

*2. Line 132: you should reference your site in Figure 1.*

**Response: We agree with this. Done.**

*3. Line 166: Long time should be long-time.*

**Response: Thank you for your suggestion. Done.**

*4. Lines 178-183: Qui et al 2019 is the referenced method for the ice phenology dataset, however, how do they account for the accuracy of the ice dataset you are using for your analysis? Using visible MODIS to ascertain freeze dates can be difficult, as the ice must be substantial enough to change the reflective properties. A few brief sentences to expand on the methods in this section would do well to provide context for the accuracy of the ice dataset you are using.*

**Response: Thank you for your insightful comments. Yes, it is important to ensure the reliability of dataset used in our paper. Actually, Qiu et al have selected six lakes (Qinghai Lake, Selin Co, Hala Lake, Dogze Co, Aksayqin, and Yaggain Co) with different locations, sizes and shapes on the QTP to verify and compare the ice coverage of this dataset and two other datasets based on passive microwave in their paper (Qiu et al., 2019). The result showed that the ice coverage obtained in their paper was highly consistent with that from passive microwave data at an average $R^2$ of 0.91 and an RMSE varying from 0.07 to 0.13 in the six lakes. And the $R^2$ and RMSE are 0.86 and 0.13, respectively in QHL, which indicates this dataset is very accurate and suitable for the division of lake ice phenology in QHL. Following your suggestion, we added the results of this data verification and comparison in this section to show that the dataset is suitable for the accuracy of our study (L202~205): This ice coverage has been compared with that from two other datasets based on passive microwave, and was found to be highly consistent with each other at an average $R^2$ of 0.86 and an RMSE of 0.13 in QHL (Qiu et al., 2019). Thus, this dataset is very accurate and suitable for the division of lake ice**

**phenology in QHL.**

**Table 2.** Parameters of linear fit between the "ice-on" proportion and lake ice coverage of this paper.

| Lake number | Lake Name | Slope | Shift | $R^2$ | RMSE |
|---|---|---|---|---|---|
| 1 | Qinghai Lake | 0.92,127 | 0.00871 | 0.86,335 | 0.13,808 |
| 2 | Serling Co | 0.9632 | 0.00077 | 0.9601 | 0.07438 |
| 3 | Hala Lake | 0.9763 | 0.00862 | 0.94,674 | 0.10,907 |
| 4 | Dogze Co | 0.93,485 | 0.00151 | 0.91,318 | 0.11,502 |
| 5 | Aksayqin | 0.90,571 | 0.0365 | 0.79,444 | 0.19,446 |
| 6 | Yaggain Co | 0.85,777 | 0.12,164 | 0.56,987 | 0.27,392 |

Figure R1: The linear fit of the ice coverage from Qiu et al. (2019) and two other datasets based on passive microwave in six lakes over QTP. This table is taken from Qiu et al. (2019).

*5. Fig S3: the x-axis should be the same for all 3 figures. They should all range from 0 to 60%; if you are to just glance at the figures and not read the axis label/units one would assume they all contribute the same during each period.*

**Response: Many thanks for your good suggestion. I think you're referring to Fig S4. And we have changed the range of x-axis to 0~60% in this revision shown in blow figure.**

[Figure]

Fig. S4. Importance of the daytime and nighttime climate factors to the evaporation (E) rate of Qinghai Lake during the ice–free and ice–covered periods (IFP and ICP). Rn, Δe. WS, WD, Pres, Ta−Ts, Tl and ICR denote the net radiation, vapor pressure difference, wind speed, wind direction, surface air pressure, difference between the air and lake surface temperatures, average temperature of the lake body from 0 to 300 cm and ice coverage rate, respectively.

*6. Fig S5: the y-axis should have the same scale for all figures. Why is the x-axis for ice cover 1 year? Whereas the IF and AN showing 3 and 4 years respectively? Your caption*

*states they are showing the results from 2014-2018.*

**Response: Thank you for your suggestion and elaborate comments. I think you're referring to Fig S6. We have unified the y-axis to be the same scale in this figure. All x-axis in this figure are the results from 2014~2018 (four years). Because the average length of AN, IFP and ICP are approximate 368, 278 and 90 days for a cycle year (AN: from the begin of IFP and the end of ICP), respectively. Thus, the sum days of AN, IFP and ICP during the four years (2014~2018) are 1472 (368×4), 1112 (278×4) and 360 (90×4) days shown as the y-axis in this figure.**

[Figure]

Fig. S6. Daily observed and simulated evaporation (E) with the atmospheric dynamics model ($E_{AD}$), mass–transfer model ($E_{MT}$) and Jensen–Haise model ($E_{JH}$) in the cycle year (annual: AN, a~c), ice–free (IF, d~f) and ice–covered (IC, h~g) periods from 2014 to 2018.

*7. Fig 1: DEM needs units, missing the line for rivers in the legend, is the scale the same for the inset map?*

**Response: Many thanks for your useful suggestion. We have added the units of DEM and line for rivers in the legend. And the scale is not the same for the inset map. The inset map is intended to show the relative position of the study area, so we did not add a scale to it. And we added a note of the scale in the Figure 1 shown as following: The scale is just for the Qinghai Lake Basin.**

[Figure]

*Figure 1. Location of Qinghai Lake (below) and the measurement site of the Chinese Torpedo Qinghai Lake test base (upper). The insets in the upper picture are photos of the four–way radiometer and infrared thermometer (left), meteorological variable measurements (middle), and eddy covariance sensors (right). The scale is just for the Qinghai Lake Basin.*

*8. When using the abbreviations for ice-covered (IC) or ice-free (IF), they are missing context (or a word) such as conditions or periods.*

**Response: Thank you for your insightful comments. To make it clear, we have changed all IC and IF to ICP and IFP as the abbreviation of ice-covered period and ice-free period in this revision.**

**References:**

Cai, Y., Ke, C. Q., Li, X., Zhang, G., Duan, Z., & Lee, H. (2019). Variations of lake ice phenology on the Tibetan Plateau from 2001 to 2017 based on MODIS data. Journal of Geophysical Research: Atmospheres, 124(2), 825–843.

Guo, Y., Zhang, Y., Ma, N., Xu, J., & Zhang, T. (2019). Long–term changes in evaporation over Siling Co Lake on the Tibetan Plateau and its impact on recent rapid lake expansion. Atmospheric research, 216, 141–150.

Kuang, X., and Jiao, J. J. (2016), Review on climate change on the Tibetan Plateau during the last half century, Journal of Geophysical Research, 121, 3979–4007.

Qiu, Y., Xie, P., Leppäranta, M., Wang, X., Lemmetyinen, J., Lin, H., & Shi, L. (2019). MODIS–based daily lake ice extent and coverage dataset for Tibetan Plateau. Big Earth Data, 3(2), 170–185.

Roderick M.L. & Farquhar, G.D. (2022). The cause of decreased pan evaporation over the past 50 years. Science 298, 1410-1411.

Wan, W., Zhao, L., Xie, H., Liu, B., Li, H., Cui, Y., ... & Hong, Y. (2018). Lake surface water temperature change over the Tibetan plateau from 2001 to 2015: A sensitive indicator of the warming climate. Geophysical Research Letters, 45(20), 11–177.

Wang, W., Lee, X., Xiao, W., Liu, S., Schultz, N., Wang, Y., ... & Zhao, L. (2018). Global lake evaporation accelerated by changes in surface energy allocation in a warmer climate. Nature Geoscience, 11(6), 410–414.

Woolway, R. I., Kraemer, B. M., Lenters, J. D., Merchant, C. J., O'Reilly, C. M., & Sharma, S. (2020). Global lake responses to climate change. Nature Reviews Earth & Environment, 1(8), 388–403

Yang, K., Wu, H., Qin, J., Lin, C., Tang, W., & Chen, Y. (2014). Recent climate changes over the Tibetan Plateau and their impacts on energy and water cycle: A review. Global and Planetary Change, 112, 79–91.

**Point to Point Response to the Reviewer#2's Comments**

*The paper is focused in exploring the key factor and salinity on E. The microclimate factors are well explored buy in regard the salinity with some weakness for two different conditions considered (IF, IC). In general, the paper provides an important technique contribution.*

**Response: Thank you very much for your positive comments. Your comments do improve our study. The main innovation of this study was quantified the E during ice-covered period (ICP) (lake ice sublimation reaches 175.22±45.98 mm, accounting for 23% of the annual evaporation) by six years of EC observations, and found that use of different models may be more reasonable to E simulate during IFP and ICP, due to the different underlying mechanism. Of course, as a saline lake, considering salinity in the models of E simulation of QHL make it more theoretical to explain E process and reduced the uncertainty of estimation. Thus, we introduced the water activity of QHL to the MT and AD models, and applied it in the MT model for E simulation of IFP during 2003 to 2017, since the JH model was chosen for the ICP. However, due to the lack of time series of lake salinity, the analysis of the difference in the effect of salinity on E between ice-free period (IFP) and ICP is relatively weak.**

**And we provide a point-to-point response to your comments in bold font below, and show amendment we made in the manuscript in underline font.**

*1. 59-61 – The paragraph includes the key and rich knowledge on saline lakes E, but is included only one author, Hamdani et al., 2018.*

**Response: Thank you for your insightful comments. We have added more relevant references in the reversion. For example, Salhotra et al. (1985) stated the effect of salinity and ionic composition on evaporation in Dead Sea; Hamdani et al (2018) and Obianyo (2019) emphasized the inhibitory effect of salinity on evaporation; and Woolway et al. (2020) reviewed the effect of climate and environmental factors on evaporation in lakes.**

*2. 117 – In this line is referred to climate changes for the period 2003-2017, why?*

*Maybe is it correct to refer only to climate variability as it written in the abstract.*

**Response: Many thanks for your good suggestion, and we changed all climate changes to be climate variability in this revision.**

*3. 121 – In title 2.1 would be better split into two chapters*

*In my opinion the Site description is poorly described, how about the other lake characteristics such as: lake topography, inflow-outflow, stratification, thermal stability, hydrodynamics?*

**Response: Thank you for your constructive comments. Following your suggestion. We first divided this section into two parts: the section of 'Study area' and 'Site description and energy exchange flux and climate data'. And we also added more information about the lake topography, inflow-outflow, and stratification (L127~134) shown as follows: Surrounded by mountains, the QHL is a typical closed tectonic depression lake, which is fed by five major rivers, including the Buha, Shaliu, Hargai, Quanji, and Heima Rivers (Jin et al., 2015). The total annual water discharge is approximately 1.56 × 10⁹ m³, of which the Buha River contributes 50% and Shaliu River contributes approximately one third (Jin et al., 2015). The mean annual Ta, precipitation, and E values between 1960 and 2015 were −0.1°C, 355 mm and 925 mm, respectively (Li et al., 2016). The seasonal stratification of QHL corresponded to that of a dimictic lake with the spring overturn taking place around May and the autumn overturn appearing around November–December (Su et al., 2019).**

*4. 128 – The sentence seems nonconclusive, it does not include any information and Reference.*

**Response: Thank you for your elaborate comments. We tried to point out the drastic changes of QHL in recent years by this sentence. Thus, to make this information more reasonable, we added two references (Tang et al., 2018; Han et al., 2021) which indicated significant increase in lake surface temperature (with a rate of 0.04 °C yr⁻¹ from 2006 to 2016) and area (with a total increase of 3% from 2006 to 2016), and decrease in ice phenology (with a rate of 0.13 days yr⁻¹ from 2000 to 2020) in this sentence.**

*5. 151 – Why the water temperature (Tl) was measured only at depth till of 3.0 m?*

**Response: Thank you for your comments. Due to the limitation of the number of temperature observation probes, the Tl was measured with five temperature probes (109 L, Campbell Scientific Inc.) at depths of 0.2, 0.5, 1.0, 2.0 and 3.0 m. Indeed, considering variation of water temperature along with depth, there is some error in replacing the lake temperature by the average temperature of 0~3 m. As we all know, the seasonal stratification of QHL corresponded to that of a dimictic lake with the spring overturn taking place around May and the autumn overturn appearing around November–December. Thus, the Tl decreases with increasing depth during ice-free period (IFP), while Tl increases with increasing depth during ice-covered period (ICP) (Figure V1). So, Tl used in this study may overestimate during IFP, and underestimate during ICP, which would increase the uncertainty in the effect of Tl on evaporation. We included this discussion in the text.**

[Figure]

Figure V1: The water temperature profile in July, August and September 2020 of Qinghai Lake (QHL). The Tl data was measured by an automatic observation system on the float at the depth of 0.5, 1.0, 3.0, 5.0, 7.0, 9.0, 11.0, 13.0 and 15.0 m with a temporal resolution of 2 hour during July to September of 2020 in QHL.

*6. 264 – To explore the key factor controlling E, was used two methods to estimate the sensitivity and the importance of each variable. Can you define the difference between*

*the two approaches, or add more literature?*

**Response: Thank you for your constructive comments and useful suggestion. Partial least squares regression is a method for linear regression between a possibly vector-valued response and a number of predictors, while random forest is a powerful tool for nonlinear regression and exploring the importance of the influence of multiple independent variables on dependent variables based on theory of decision tree. Thus, partial least squares regression and random forest were used to analyze the relationship between E and climate and environmental factors from linear and nonlinear processes, respectively, which has been widely used in the study of hydrological and ecological field. Following your suggestion, we added some more description and literature about the two approaches in this revision shown as following: The two methods analyze the relationship between E and climate and environmental factors from linear and nonlinear processes, respectively, and have been widely used in the study of hydrological and ecological fields (Desai and Ouarda, 2021; Li et al., 2022; Sow et al., 2022).**

*7. 362-385 - Was discussed the influence of salinity on E rate. Then E-IF and E-IC rate within the saline lake environment in consequence is one of the interests. I think the literature explains quite enough the salinity influence on E, but this paper attends to describes the within the two different thermal conditions and no results. Is explained that was measured the water activity by 0.97 and applied to the model, that is it (IF, IC ?).*

**Response: Thank you for your elaborate comments. Yes, this paragraph was used to discuss the application of water activity by 0.97 in the models during IF. Although the application of water activity has little effect on the evaporation value of QHH (a decrease of 3% and approximate 24 mm/yr), we think that it is important to consider the effect of salinity on the evaporation simulation for the explanation of the mechanism in models of saline lakes. Because this influence increases gradually over saline. When the salinity concentrations are 100 g $L^{-1}$ and 300 g $L^{-1}$, the reductions in evaporation are 3.4% and 31.9%, respectively. Thus, it is more reasonable to consider the effect of salinity on evaporation in saline lakes.**

Following your suggestion, we clarified our expression in the section of '2.6. Models for daily lake evaporation simulation' (L236~248) as following: **Considering that Qinghai Lake is a saline lake, and many studies have pointed out that it is valuable to consider the influence of salinity on saline lake evaporation, and with the increase of salinity, it will exert greater inhibition on evaporation (Hamdani et al., 2018; Mor et al., 2018). Thus, the water activity coefficient (α) which is defined as the ratio between the vapor pressure above saline water and that above freshwater at the same temperature has been introduced to characterize the effect of salinity on saline lake evaporation (Salhotra et al., 1987; Lensky et al., 2018). Because saline water drains out salt during freezing (Badawy, 2016), we only introduced the α into the evaporation simulation of Qinghai Lake during IFP.**

we discussed it at the same time as following: **In our study, we measured the water activity of QHL as 0.97 by a salinity of 14.13 g L$^{-1}$, and applied it to the MT and AD models for E simulation of IFP during 2003 to 2017, which make it more theoretical to explain the E process of saline lakes and reduced the uncertainty of estimation in saline lake E.**

And we also divided this long paragraph into two parts of '4.2. Responses of lake evaporation to salinity' and '4.3. Responses of lake evaporation to climate variability' in this revision.

*9. In the same long paragraph, is defined the reduced saturated vapor pressure above the water (at a given water temperature, which one?); in other wise, there must be more evaporation, but here is given the opposite definition. This section must be clarified.*

**Response: Thank you very much for your constructive comments. Actually, the water activity (α) is defined as the ratio of water vapor pressure on the surface of saline and fresh water at the same temperature, which can be shown as following equation:**

$$\alpha = \frac{e_{ss}(Ts)}{e_{sf}(Ts)}$$

**Where $e_{ss}$ and $e_{sf}$ are the water vapor pressure at the same temperature (Ts),**

respectively. Because $e_{ss}$ is smaller than $e_{sf}$ at the same temperature, $\alpha$ always less than 1, and $\Delta e$ would be smaller in saline lakes than that in fresh lakes at the same temperature. Thus, there is less evaporation in saline lakes than that in fresh. The statement is easy to lead to misunderstand, so we have replaced the 'a given temperature' with 'the same temperature' in this reversion.

10. Dear Editor,

Can you provide the list of paper:

Hamdani et al., 2018

Salhotra et al., 1987

Wang et al., 2019[a]

Response: We will upload these articles along with our response.

**References:**

Salhotra, A. M., Adams, E. E., & Harleman, D. R. (1985). Effect of Salinity and Ionic Composition on Evaporation: Analysis of Dead Sea Evaporation Pans. Water Resources Research, 21(9), 1336–1344.

Hamdani, I., Assouline, S., Tanny, J., Lensky, I. M., Gertman, I., Mor, Z., & Lensky, N. G. (2018). Seasonal and diurnal evaporation from a deep hypersaline lake: The Dead Sea as a case study. Journal of Hydrology, 562, 155–167.

Obianyo, J. I. (2019). Effect of Salinity on Evaporation and the Water Cycle. EmergingScience Journal, 3(4): 256–262.

Woolway, R. I., Kraemer, B. M., Lenters, J. D., Merchant, C. J., O'Reilly, C. M., & Sharma, S. (2020). Global lake responses to climate change. Nature Reviews Earth & Environment, 1(8), 388–403.

Jin, Z. D., An, Z. S., Yu, J. M., Li, F. C., & Zhang, F. (2015). Lake Qinghai sediment geochemistry linked to hydroclimate variability since the last glacial. Quaternary Science Reviews, 122(2015): 63–73.

Li, X. Y., Ma, Y. J., Huang, Y. M., Hu, X., Wu, X. C., Wang, P., ... & Liu, L. (2016). Evaporation and surface energy budget over the largest high-altitude saline lake on the Qinghai-Tibet Plateau. Journal of Geophysical Research: Atmospheres, 121(18), 10–470.

Li, X. Y., Shi, F. Z., Ma, Y. J., Zhao, S. J., & Wei, J. Q. (2022). Significant winter $CO_2$ uptake by saline lakes on the Qinghai–Tibet Plateau. Global Change Biology, 2022, 28(6): 2041–2052.

Su, D. S., Hu, X. Q., Wen, L. J., Lyu, S. H., Gao, X. Q., Zhao, L., … & Kirillin, G. (2019). Numerical study on the response of the largest lake in China to climate change. Hydrology and Earth System Sciences, 23: 2093–2109.

Tang, L. Y., Duan, X. F., Kong, F. J., Zhang, F., Zheng, Y. F., Li, Z., … & Hu, S. J. (2018). Influences of climate change on area variation of Qinghai Lake on Qinghai–Tibetan P Han, W. X., Huang, C. L., Gu, J., Hou, J. L., & Zhang, Y. (2021). Spatial–Temporal Distribution of the Freeze–Thaw Cycle of the Largest Lake (Qinghai Lake) in China Based on Machine Learning and MODIS from 2000 to 2020. Remote Sensing, 13(9): 1695.lateau since 1980s. Scientific Report, 8: 7331–7338.

Desai, S., & Ouarda, T. B. M. J. (2021). Regional hydrological frequency analysis at ungauged sites

with random. Journal of Hydrology, 594: 125861.

Sow, A., Traore, I., Diallo, T., Traore, M., & Ba, A. (2022). Comparison of Gaussian process regression, partial least squares, random forest and support vector machines for a near infrared calibration of paracetamol samples. Results in Chemistry, 4: 100508

---

## Referee Report (RR1)

In this study, the authors quantified evaporation/sublimation (E) during ice-free and ice-cover periods (IFP and ICP) for a large lake on the Tibetan Plateau. Field observations were collected between 2014 to 2019 and used to quantify evaporation/sublimation (E) and then used to determine the main controls on E during the IF and IC period and annually. The results highlight a vast E (23%) during ICP in Qinghai Lake, and revealed the differences in controlling factors of E during IFP and ICP. This manuscript remains a very interesting piece of research, which provides important new insights by 6 years of direct observation. My previous comments/concerns have been adequately addressed with additional analysis and explanation by the authors, and as such the results and conclusions of the new version are more convincing. A few relatively minor issues remain but I believe that these should be able to be addressed without the need for further review.

**Specific comments:**

1) Line 41, '2003–2017' may be '2003~2017' as shown in Line124, 314, 315 and so on.

2) Line 104, '0.037°C/yr' should be '0.037 °C/yr'.

3) Line 131, '−0.1°C' should be '−0.1 °C'.

4) Line 135, I do not think it is precise to claim that the length of ICP is more than 100 days in QHL. The length of ICP was 83~121 days during 2002~2014 (Fig. S4 in this study), with only a very few years exceeding 100 days. Thus, 'lasts more than 100 days' is suggested to be 'lasts approximate 100 days'.

5) Line 338, 'This indicated that E of QHLwas mainly controlled by WS' should be 'This indicated that E of QHL was mainly controlled by WS'.

6) Line 348, '−0.01°C/yr' should be '−0.01 °C/yr'. The authors need to carefully check the format of the full text, space is needed between numbers and units.

7) Line 367, '0.73–1.38' may be '0.73~1.38'.

8) Line 404, '3h$^{-1}$' should be '3 h$^{-1}$'.

9) Line 485, 'reduction of 11.1%' should be 'reduction of 7.56%'.

10) Considering the difference in the driving factors of lake evaporation during IFP and ICP, it should be very cautious in year-round E simulation by the traditional formula of E, before a reasonable verification in different seasons. One speculative comment is

whether the Penman formula series considering both aerodynamics and energy balance would work well for evaporation simulations during both IFP and ICP, although the parameter input for this model may be a bit more complex. I think this is worth exploring in the future study.

---

## Author Response (AR2)

Dear Editor and Reviewers,

We appreciate very much the valuable and constructive comments on our manuscript entitled "Evaporation and sublimation measurement and modelling of an alpine saline lake influenced by freeze–thaw on the Qinghai–Tibet Plateau" (ID hess-2023-100). We have carefully revised the manuscript according to your comments. The following paragraphs respond to the specific comments of referees, the original review comments are listed first in their originals (in italic), followed by our itemized responses. All line numbers listed in this response refer to the manuscript with tracks. Hope this revision can adequately address comments raised by referees.

Best regards,

Sincerely yours,

Authors of the Manuscript

*Point to Point responses to Reviewer's Comments*

➢ *Reviewer: 1*

*In this study, the authors quantified evaporation/sublimation (E) during ice-free and ice cover periods (IFP and ICP) for a large lake on the Tibetan Plateau. Field observations were collected between 2014 to 2019 and used to quantify evaporation/sublimation (E) and then used to determine the main controls on E during the IF and IC period and annually. The results highlight a vast E (23%) during ICP in Qinghai Lake, and revealed the differences in controlling factors of E during IFP and ICP. This manuscript remains a very interesting piece of research, which provides important new insights by 6 years of direct observation. My previous comments/concerns have been adequately addressed with additional analysis and explanation by the authors, and as such the results and conclusions of the new version are more convincing. A few relatively minor issues remain but I believe that these should be able to be addressed without the need for further review.*

**Response: Many thanks for your positive and constructive comments. We have carefully checked the overall manuscript and corrected many format errors.**

*Specific comments:*

*1) Line 41, '2003–2017' may be '2003~2017' as shown in Line124, 314, 315 and so on.*

**Response: Many thanks! Yes, we have corrected it in this revision.**

*2) Line 104, '0.037°C/yr' should be '0.037 °C/yr'.*

**Response: Thanks a lot, done.**

*3) Line 131, '−0.1°C' should be '−0.1 °C'.*

**Response: Thank you very much, done.**

*4) Line 135, I do not think it is precise to claim that the length of ICP is more than 100 days in QHL. The length of ICP was 83~121 days during 2002~2014 (Fig. S4 in this study), with only a very few years exceeding 100 days. Thus, 'lasts more than 100 days' is suggested to be 'lasts approximate 100 days'.*

**Response: Thank you for pointing out this. We agree with this and corrected as: 'lasts approximate 100 days' in Line 137.**

*5) Line 338, 'This indicated that E of QHLwas mainly controlled by WS' should be 'This indicated that E of QHL was mainly controlled by WS'.*

**Response: Many thanks! We revised this sentence as suggested in Line 342.**

*6) Line 348, '−0.01°C/yr' should be '−0.01 °C/yr'. The authors need to carefully check the format of the full text, space is needed between numbers and units.*

**Response: Many thanks! We have carefully checked the overall manuscript and corrected many format errors.**

*7) Line 367, '0.73–1.38' may be '0.73~1.38'.*

**Response: We have corrected them, thank you!**

*8) Line 404, '3h−1' should be '3 h−1'.*

**Response: Done.**

*9) Line 485, 'reduction of 11.1%' should be 'reduction of 7.56%'.*

**Response: Thanks for your careful suggestion. The value '11.1%' is a miswriting of '7.56%', and we have corrected it in this revision in Line 490.**

*10) Considering the difference in the driving factors of lake evaporation during IFP and ICP, it should be very cautious in year-round E simulation by the traditional formula of E, before a reasonable verification in different seasons. One speculative comment is whether the Penman formula series considering both aerodynamics and energy balance would work well for evaporation simulations during both IFP and ICP, although the parameter input for this model may be a bit more complex. I think this is worth exploring in the future study.*

**Response: Thank you again for your constructive comments. Yes, we agreed with your point. The energy-budget-based methods (Such as the Bowen ratio energy budget, Penman, Priestley-Taylor, Brutsaert-Stricker, DeBruin-Keijman and lake thermodynamics methods) are the best choice for lake evaporation simulation when heat storage in the water can be estimated accurately (Wang et al., 2019), which indicates the observation and simulation of the thermodynamics of the vertical gradient of lakes are very important for the accurate estimation of lake evaporation. Considering the complexity of model parameters, Penman formula series was not used in this study.**

**In addition, as in your previous comments, the main highlight of this study is the finding of a vast E (23%) during ICP in Qinghai Lake and the differences in controlling factors of E during IFP and ICP by 6 year-round continuous EC observation. Hereby, we further concentrated on verifying the consistency of the accuracy of the traditional models for the evaporation simulation during ice-free periods and ice-covered periods, because almost all models were**

calibrated and verified against evaporation observations during the ice-free periods, while evaporation (or sublimation) during the ice-covered periods was either not calculated or unverified.

Moreover, as your suggestion, we are exploring the applicability of different types of models to simulate lake evaporation over the Tibetan Plateau in our next study (Figure R1) and are designing the observation system of lake thermodynamics parameters, such as sampling of lake ice in ice-covered periods (Figure R2), and verify and develop a suitable 1D or even 3D lake thermodynamics evaporation models for Qinghai Lake (or even lakes in the Qinghai-Tibet Plateau) in the future study.

[Figure]

Figure R1. Comparison in observed ($E_O$) and simulated evaporation obtained using the calibrated methods of atmospheric dynamics ($E_{AD}$), Bowen ratio ($E_{BW}$), Priestley-Taylor (EPM) and mass–transfer ($E_{MT}$) in Qinghai Lake, Siling Co and Ngoring.

[Figure]

FigR2. Sampling and measurement of lake ice of Qinghai Lake in Feb 2023.

References:

Wang, B., Ma, Y., Ma, W., Su, B., & Dong, X. 2019. Evaluation of ten methods for estimating evaporation in a small high-elevation lake on the Tibetan Plateau. Theoretical and applied climatology, 136, 1033-1045.